# Highly efficient octave-spanning long-wavelength infrared generation with a 74% quantum efficiency in a $\chi^{(2)}$ waveguide

Bo Hu[1,6], Xuemei Yang[1,6], Jiangen Wu[2,6], Siyi Lu[1], Hang Yang[1], Zhe Long[1], Linzhen He[1], Xing Luo[3], Kan Tian[1], Weizhe Wang[1], Yang Li[1], Han Wu ⬡[1] ✉, Wenlong Li[4], Chunyu Guo[3], Huan Yang ⬡[2] ✉, Qi Jie Wang ⬡[5] & Houkun Liang ⬡[1] ✉

The realization of compact and efficient broadband mid-infrared (MIR) lasers has enormous impacts in promoting MIR spectroscopy for various important applications. A number of well-designed waveguide platforms have been demonstrated for MIR supercontinuum and frequency comb generations based on cubic nonlinearities, but unfortunately third-order nonlinear response is inherently weak. Here, we propose and demonstrate for the first time a $\chi^{(2)}$ micrometer waveguide platform based on birefringence phase matching for long-wavelength infrared (LWIR) laser generation with a high quantum efficiency. In a ZnGeP$_2$-based waveguide platform, an octave-spanning spectrum covering 5–11 μm is generated through optical parametric generation (OPG). A quantum conversion efficiency of 74% as a new record in LWIR single-pass parametric processes is achieved. The threshold energy is measured as ~616 pJ, reduced by more than 1-order of magnitude as compared to those of MIR OPGs in bulk media. Our prototype micro-waveguide platform could be extended to other $\chi^{(2)}$ birefringence crystals and trigger new frontiers of MIR integrated nonlinear photonics.

In the recent decade, tremendous efforts have been spent towards the generation of ultra-broadband mid-infrared (MIR, 3–20 μm) laser sources for various applications including frequency metrology and trace-gas sensing[1]. Among various approaches, quantum cascaded lasers show good promise in device compactness. However, quantum cascaded lasers still face limitations in producing ultra-broadband emissions and ultrashort pulses, particularly in the femtosecond time scale. Up to now, nonlinear frequency conversions have become the main route for generating coherent ultra-broadband MIR radiations, which however generally consist of large and complicated laser apparatuses and exhibit relatively low conversion efficiency[2,3].

An unremitting pursuit marching toward compact and efficient MIR conversions has led to substantial progress of MIR laser sources towards on-chip integrated photonic devices. Integrated MIR emitters based on cubic polarizations ($\chi^{(3)}$) namely Kerr nonlinearities have been intensively investigated in material systems such as silicon (Si), silicon nitride (SiN), germanium (Ge) and chalcogenide leveraging on the advancement of semiconductor and CMOS technologies[4–11]. Broadband MIR supercontinuum[9,10] with the wavelength extending up to 13 μm in SiGe waveguides and frequency combs[4,5] in Si micro-resonators have been demonstrated. However, $\chi^{(3)}$ nonlinear response is inherently weak. Hence, dedicated

[1]School of Electronics and Information Engineering, Sichuan University, 610064 Chengdu, Sichuan, China. [2]Sino-German College of Intelligent Manufacturing, Shenzhen Technology University, 518118 Shenzhen, Guangdong, China. [3]College of Physics and Optoelectronic Engineering, Shenzhen University, 518060 Shenzhen, China. [4]Chengdu Dien PHOTOELECTRIC Technology Co., Ltd., 610100 Chengdu, Sichuan, China. [5]School of Electrical & Electronic Engineering & The Photonics Institute, Nanyang Technological University, Singapore 639798, Singapore. [6]These authors contributed equally: Bo Hu, Xuemei Yang, Jiangen Wu. ✉e-mail: hanwu@scu.edu.cn; yanghuan@sztu.edu.cn; hkliang@scu.edu.cn

dispersion engineering, moderate-to-small effective mode area, and resonators with high quality factors are usually required to achieve a reasonable conversion efficiency[4–6,9–11]. On the other hand, more efficient quadratic nonlinearity ($\chi^{(2)}$)-based waveguide devices for parametric conversions such as optical parametric generation/amplification (OPG/OPA), and difference-frequency generation (DFG) are expected to be a promising alternative approach for simple and efficient generation of ultra-broadband MIR lasers.

$\chi^{(2)}$-based waveguide devices have been realized empowered by quasi-phase matching technique[12–15]. OPG emitting in the wavelength range of 1700–2700 nm has been demonstrated with a moderate quantum conversion efficiency of 22.2% and a parametric gain of 118 dB/cm in a nano-waveguide platform based on periodically-poled lithium niobate (PPLN)[12]. However, the transmission window of PPLN waveguide platforms imposes restrictions on endeavor based on it for efficient long-wavelength infrared (LWIR) laser generation and application. In order to extend the wavelength into the LWIR region, particularly beyond 5 μm, the orientation-patterned gallium arsenide (OP-GaAs) platform has been employed for an on-chip OPG demonstration, which is the only $\chi^{(2)}$ waveguide device so far demonstrated in the LWIR region[15]. Nevertheless, high efficiency of parametric conversion or broadband emission has not been demonstrated in OP-GaAs waveguide platforms, and sophisticated fabrication procedures are also required for the necessity of orientational patterning, which poses stringent constraints on the selection of nonlinear media. In addition, dry-etching techniques such as inductively coupled plasma etching are made use for the fabrication of OP-GaAs waveguides, which restricts the cross-sectional area of the integrated device. Therefore, there is vital urgency and importance to unlock new $\chi^{(2)}$ waveguide platforms for simple and highly-efficient generation of LWIR laser emission with other phase-matching (PM) techniques.

In this work, we, for the first time to the best of our knowledge, propose and demonstrate the $\chi^{(2)}$ parametric micro-waveguide platform with birefringence PM for highly efficient single-pass LWIR generation, taking the advantages of high birefringence $\chi^{(2)}$ in non-oxide nonlinear crystals, such as $ZnGeP_2$ (ZGP), $AgGsS_2$, GaSe and $CdSiP_2$ which are attractive for broadband LWIR generation[16–19]. OPG in the LWIR region from the $\chi^{(2)}$ waveguide device is experimentally exploited, driven at a central wavelength of 2.4 μm. The generated idler pulse has an octave spectrum spanning from 5 to 11 μm. Owing to the inherently efficient quadratic nonlinear response, waveguide-enhanced spatial confinement and elongated interaction length, and good coupling efficiency benefited from multi-micrometer dimension design in the $\chi^{(2)}$ waveguide with birefringence PM, a low threshold pulse energy and peak power are demonstrated as ~616 pJ and 1.9 kW, respectively. More strikingly, a quantum conversion efficiency of 74% is achieved as a new record of LWIR single-pass parametric processes, which is enhanced by more than twofolds compared to the state-of-art single-pass parametric conversions in bulk crystals in the LWIR region. The saturated parametric gain of >58.6 dB/cm is observed at a pump energy of 7.9 nJ. Besides the remarkable laser specifications, the micro-waveguide device fabrication adopts a simple and effective device fabrication technique, namely bonding and grinding the ZGP nonlinear crystal with a designed PM angle, followed by patterning using ultrafast laser direct writing (ULDW) technique, which provides a simple, universal and flexible fabrication method for nonlinear micro-photonic devices. The demonstrated prototype micro-waveguide platform could be extended to other $\chi^{(2)}$ birefringence crystals for highly efficient and broadband tunable MIR laser generation. This work opens an exciting and simple route towards to the portable or on-chip MIR nonlinear photonics and practical applications of MIR spectroscopy and metrology.

## Results

### $\chi^{(2)}$ waveguide device with birefringence PM: design and fabrication

A ridge nonlinear micro-waveguide is designed for the proof-of-concept demonstration of the integrated $\chi^{(2)}$ device with birefringence PM. The ZGP crystal is chosen as the material platform for its high $\chi^{(2)}$ nonlinearities ($d_{36}$-75 pm/V), mature growth technique, and broad MIR transparency window (~0.73–12 μm)[16]. In Supplementary Note 1, detailed information of reasons why taking ZGP crystal as an example of waveguide media is provided. Traditionally, crystal angle is twisted to fulfil the birefringence PM condition. Thus, in waveguide devices with birefringence PM, with a fixed crystal angle, it is crucial to have an ultrabroad PM bandwidth such that different pump and signal wavelengths could be adapted, and the spectral tunable parametric conversion could be realized. Fortunately, in a bulk ZGP crystal, Type-I PM pumped at ~2.4 μm could provide an ultra-broadband PM bandwidth for an idler wavelength spanning from 5 to 11 μm, as calculated in Fig. 1a, with a PM angle designed as 48.3°. Besides the ultrabroad PM bandwidth, a multi-wavelength-scale ZGP ridge waveguide structure on fused silica ($SiO_2$) substrate is designed for the balance of tight field confinement, mode mismatch, and pump alignment tolerance. In the simulation of electric field distribution in the $\chi^{(2)}$ waveguide, the waveguide dimensions are designed as 35 and 40 μm in width and height, respectively. In addition, a sidewall angle of 85° is also considered in the simulation to mimic the actual fabricated device. The moderate waveguide dimension and high refractive index contrast between the ZGP waveguide core and cladding layers ($\Delta n$ ~ 2.14 for ZGP versus air or $\Delta n$ ~ 1.71 for ZGP versus $SiO_2$) contribute to an excellent mode confinement. As illustrated in simulated mode cross sections in Fig. 1b, fundamental mode profiles of the 2.4 μm pump and 8 μm idler in transverse magnetic ($TM_{00}$) and electric ($TE_{00}$) polarizations, corresponding to ordinary and extraordinary waves, respectively, of ZGP crystal are well confined and overlapped in the $\chi^{(2)}$ waveguide. Notably, confinement factors of both $TE_{00}$ and $TM_{00}$ modes are calculated to be greater than 99% in the entire interested LWIR spectral range. Moreover, the propagation loss of fundamental modes including the material absorption and anisotropy related mode leakage[20,21] in the designed $\chi^{(2)}$ waveguide in a wavelength range of 2 to 11 μm are calculated, as presented in Fig. 1c. The propagation loss of fundamental TE and TM mode is less than 1.5 dB/cm and 4.0 dB/cm, respectively. Notably, it is revealed that the pump wave at 2.4 μm in $TM_{00}$ mode has a propagation loss <0.01 dB/cm, and losses of fundamental modes are <0.05 dB/cm at 8 μm, and <0.25 dB/cm (TE) and 0.5 dB/cm (TM) at 9 μm, respectively, which guarantees a good transmission of pump and parametric waves in the $\chi^{(2)}$ waveguide. The raise of waveguide losses in the wavelength range of 9–11 μm is attributed to the absorption of fused silica substrate peaked at 9.5 μm, as shown in the top inset which depicts the absorption coefficient of silica, as a function of wavelength[22]. This implies that substrates such as sapphire with low absorption in the LWIR region could further improve the performance of the nonlinear waveguide. It is worth mentioning that the material loss of ZGP crystal is minimal in the spectral range of 2–11 μm as shown by the measured transmission spectrum of a 10-mm-thick uncoated ZGP crystal in the bottom inset of Fig. 1c (The Fresnel reflection is not subtracted).

Evaluation of dispersion is critical for the design of $\chi^{(2)}$ waveguide devices. The group velocity dispersion (GVD) and group velocity mismatch (GVM) between pump and idler pulses in the $\chi^{(2)}$ waveguide are calculated and presented in Fig. 1d. Owing to the multi-wavelength-scale geometry, the effect of waveguide dispersion is weak, and thus the ZGP waveguide features a nearly identical GVD profile with that of the bulk material, which avoids dedicated dispersion engineering and hence simplifies the design of $\chi^{(2)}$ waveguide devices with birefringence PM. In addition, the GVD of 2.4 μm pump pulse is calculated as a small

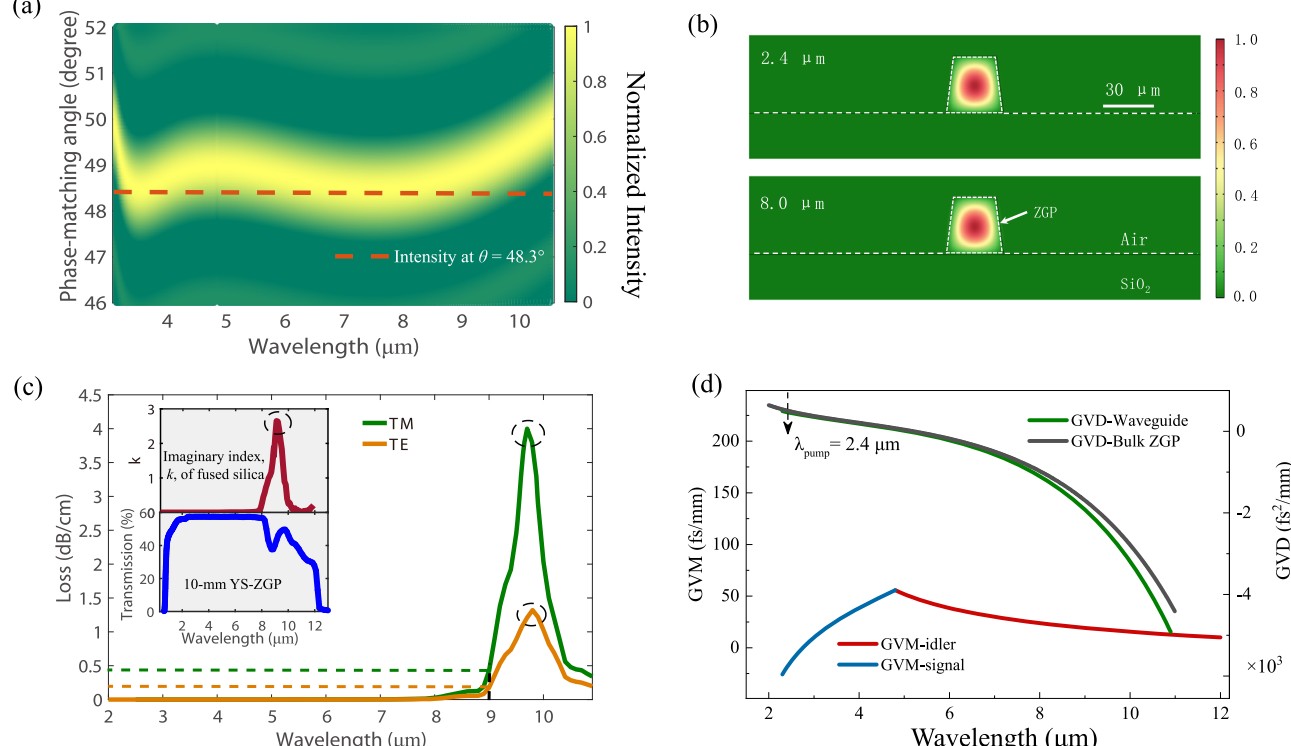

**Fig. 1 | Design and simulation of the $\chi^{(2)}$ waveguide with birefringence PM based on a ZGP crystal. a** Calculated normalized gain spectrum at a pump wavelength of 2.4 μm for a bulk ZnGeP$_2$ (ZGP) crystal, at various phase match (PM) angles. A PM angle of $\theta$ = 48.3° is designed as marked by the orange dashed line to generate the broadest gain bandwidth. **b** Simulated electric field distributions in the $\chi^{(2)}$ wave-guide of pump wave at 2.4 μm in TM$_{00}$ mode (top) and a typical idler wave at 8.0 μm in TE$_{00}$ mode (bottom). **c** Calculated propagation losses of fundamental modes in the designed $\chi^{(2)}$ waveguide in TE and TM polarizations with the angle between the propagation direction and the crystal optical axis as 48.3°, in a broad spectral range of 2–11 μm. The results indicate that the propagation loss of fundamental TE and TM mode is less than 1.5 dB/cm and 4.0 dB/cm, respectively. Notably, calcu-lated propagation losses of fundamental modes are <0.05 dB/cm at 8 μm, and

<0.25 dB/cm (TE) and 0.5 dB/cm (TM) at 9 μm, respectively. The raise of waveguide losses in the wavelength range of 9–11 μm is attributed to the absorption of fused silica substrate peaked at 9.5 μm, as shown in the top inset which depicts the absorption coefficient (imaginary part of complex refractive index) of silica, as a function of wavelength[22]. Meanwhile, the material loss of ZGP crystal is minimal in the spectral range of 2–11 μm as shown by the measured transmission spectrum of a 10-mm-thick uncoated ZGP crystal in the bottom inset of **c**. **d** The simulated group velocity dispersion (GVD) of TM$_{00}$ modes in a wavelength range of 2 to 11 μm, in the designed $\chi^{(2)}$ ZGP waveguide (green) and bulk ZGP (black), respectively. Owing to multi-wavelength-scale dimensions, the nonlinear waveguide exhibits nearly an identical GVD profile to that of the bulk material, which simplifies the design of $\chi^{(2)}$ waveguide devices with birefringence PM.

value of 470 fs$^2$/mm, which implies that the temporal profile of the pump pulse is not obviously perturbed with respect to the 320 fs pulse width, while propagating in the 10-mm-long $\chi^{(2)}$ waveguide. On the other hand, the calculated maximum GVM between the pump at a wavelength of 2.4 μm and the idler spanning from 6 to 12 μm is less than 30 fs/mm, indicating a small temporal walk off for parametric waves propagating in the 10-mm-long ZGP waveguide.

Based on the designed parameters, a simple and effective micro-waveguide fabrication technique which combines procedures of nonlinear crystal bonding with the designed PM angle, $\chi^{(2)}$ wafer lap-ping and ULDW is employed and demonstrated based on a ZGP crystal, as depicted in Fig. 2. As the first step, a bulk ZGP crystal (DPT, YS-ZGP) with the dimension of 4 (width) × 3 (thickness) × 10 (length) mm$^3$ is cut into a crystal angle of 48.3° for type I PM. The bulk crystal is then bonded to a fused silica substrate using ultraviolet curing optical adhesives. The bonded bulk ZGP crystal is subsequently lapped and polished to a thickness of ~40 μm. Figure 2g shows the photograph of lapped ZGP-on-SiO$_2$ wafer. Finally, the ULDW technique is adopted to form a ridge micro-waveguide structure on the lapped ZGP wafer (see Fig. 2e, f). More detailed descriptions of the ULDW process are presented in Methods. Scanning electron microscope images of the fabricated ZGP ridge waveguides based on the proposed fabrication technique for $\chi^{(2)}$ devices are also displayed in Fig. 2h. It is measured that the structured ZGP waveguide exhibits a side-wall roughness <0.6 μm (see detailed information in Supplementary Note 5), and the

waveguide facets are nearly identical to those of unprocessed ZGP films, indicating a small scattering loss and potentially good coupling efficiency of the fabricated $\chi^{(2)}$ waveguide. In addition, the transmission loss of fabricated waveguide at 2.4 μm pump wavelength in TM polarization is also measured as 0.8 dB/cm (Supplementary Note 5), indicating a good waveguide performance for the short nonlinear device. The discrepancy between the measured and simulated trans-mission loss is attributed to the wave scattering by the waveguide surface and side walls. Notably, ULDW with optimized processing parameters is effective for etching high-quality structures with tens of micrometer depth in $\chi^{(2)}$ nonlinear crystals, with minor constraint on materials to be processed, which provides a unique merit compared to traditional CMOS dry-etching techniques such as inductively coupled plasma etching. It is therefore suggested that adopted technique could be a universal method for large-scale three-dimensional fabrication of $\chi^{(2)}$ micro-waveguide device.

## Highly efficient and low threshold parametric generation

To characterize the parametric conversion in the fabricated $\chi^{(2)}$ micro-waveguide device with birefringence PM, OPG is performed to mea-sure the parametric conversion efficiency, parametric gain bandwidth and saturated gain value[12,13] with the experimental setup illustrated in Fig. 3a. The pump source is a home-built OPA centered at 2.4 μm with 320 fs pulse width at a repetition rate of 500 kHz. The spectrum of the pump laser is presented in Fig. 3b (More details of the MIR OPA as the

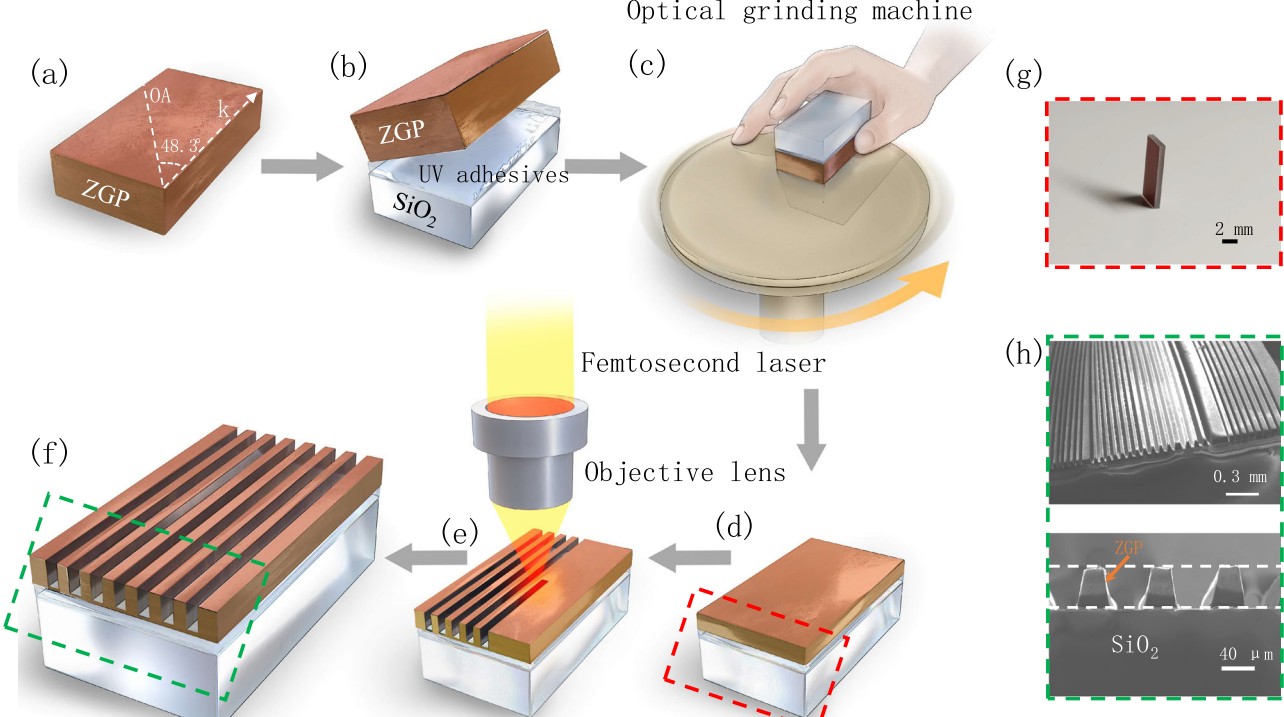

**Fig. 2 | Fabrication steps of the $\chi^{(2)}$ waveguide with birefringence PM based on a ZGP crystal. a, b** Step1**:** a bulk ZGP crystal with a PM angle of 48.3° for type I PM (the designed optical axis (OA), with respect to the incident wave vector **k**, is marked by a white dashed line) is bonded to a fused silica substrate, through using the ultraviolet curing optical adhesives. **c, d** Step 2**:** the bonded bulk ZGP crystal with the designed PM angle is lapped and polished to a thickness of ~40 μm by employing an optical grinding machine. **e, f** Step 3**:** the ultrafast laser direct writing technique is adopted to form a ridge waveguide structure on the lapped ZGP wafer. **g** The photograph shows the grinded ZGP-on-SiO₂ wafer. **h** Scanning electron microscope images of the fabricated $\chi^{(2)}$ waveguide. Closely packed waveguide arrays are fabricated to fully utilize the $\chi^{(2)}$ wafer, while no field could be coupled between the adjacent waveguides.

pump source are included in "Methods."). The 2.4 μm pump beam is coupled into the ZGP waveguide focused by an uncoated CaF₂ lens with a 40-mm focal length, while the generated LWIR idler is collected by a ZnSe lens with a 25-mm focal length and an anti-reflection coating in the spectral range of 2–13 μm. An input coupling efficiency of 15.3% and an output collecting efficiency of 34% are measured and estimated, respectively (More details of the measurement and calculation of coupling efficiencies are presented in "Methods."). A long-pass filter with a cutoff wavelength of 4.5 μm is used to remove the residual co-propagating pump and signal beams, and a MIR hollow core fiber with a 500 μm core diameter is employed to transmit the generated LWIR radiation. The spectrum of the generated idler is characterized by using a grating-scanning monochromator equipped with a lock-in amplifier and a liquid nitrogen cooled Mercury Cadmium Telluride (MCT) detector. The average power of the generated idler waves is measured using the MCT detector and a power meter (More detailed information of employed measurement equipment is placed at "Methods.").

Measured OPG spectra at different pump energy (peak power) are shown in Fig. 3c. Driven at a pulse energy of 3.2 nJ, corresponding to a peak power of 9.4 kW, the OPG spectrum spans from 6.6 to 9.6 μm. As the pump energy is increased to 8.2 nJ, a macroscopic broadening of OPG spectra is observed because of the parametric gain bandwidth broadening[12] which could tolerate certain deviation of the PM angle from the designed value caused by fabrication imperfections. Notably, the generated LWIR idler wave has an octave-spanning spectrum covering 5.4 to 10.3 μm at −20 dB, which is in good agreement with the simulated result in Fig. 1a. The output power of the generated LWIR idler from the ZGP waveguide is plotted in Fig. 3d, as a function of pump pulse energy and average power (The corresponding signal information could be found in Supplementary Note 2). An OPG

threshold of 616 pJ is measured from the ZGP waveguide, which is reduced by more than 1-order of magnitude, compared to the state-of-the-art MIR single-pass OPGs based on bulk crystals. It is worth mentioning that the threshold measurement is limited by the transmission loss of ZnSe lenses and LPF, the noise floor of the used MCT detector and 1-s integration time of the lock-in detection. Even lower threshold energy value is expected with a smaller waveguide dimension and corresponding dispersion engineering. Increasing the pump pulse energy, the OPG output power grows exponentially. The LWIR OPG power is measured as ~0.72 mW, at a pump pulse energy of 6.5 nJ and an average power of 3.25 mW, as presented in Fig. 3d, corresponding to a power efficiency of 22% and a quantum efficiency of 74%, which is a record high value for single-pass optical parametric conversion in LWIR. Moreover, as shown in Fig. 3e, the measured saturated gain of the nonlinear waveguide is ~58.6 dB/cm, which indicates that a decent parametric gain could be provided by the $\chi^{(2)}$ waveguide device[12,13]. The inset of Fig. 3e is the measured beam profile of the LWIR idler from the ZGP waveguide. The revealed spatial intensity distribution is in excellent agreement with the simulated mode profile shown in Fig. 1b.

The OPG spectral tunability from the $\chi^{(2)}$ waveguide with birefringence PM is investigated by scanning the pump wavelength in the range of 2.3–2.7 μm. As shown in Fig. 4a, pumped at 2.4 μm, a LWIR idler spectrum centered at 8 μm is obtained. When the pump wavelength is increased to 2.5 μm, owing to better PM condition at two wings of the spectrum, an octave-spanning idler covering 4.5 to 12 μm is generated. Further increasing the pump wavelength to 2.6 μm abruptly reduces the parametric bandwidth, and a relatively narrow-band idler spectrum centered at 9.3 μm is measured, as presented in Fig. 4c. A semi-classical simulation of OPG is also conducted to study the spectral evolution in the $\chi^{(2)}$ waveguide with birefringence PM at different pump wavelengths, by solving the $\chi^{(2)}$-based coupled-wave

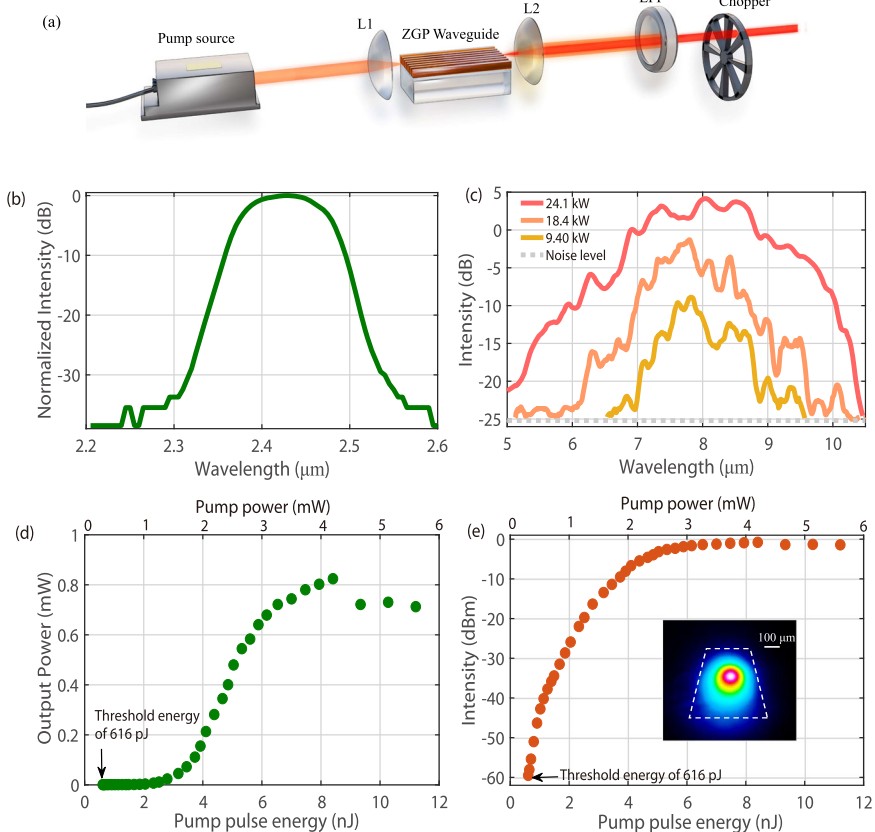

**Fig. 3 | Experimental setup and characterizations of the spectra, output power, pump threshold and parametric gain of the $\chi^{(2)}$ micro-waveguide with birefringence PM based on a ZGP crystal. a** The schematic of the experimental setup for optical parametric generation (OPG) from the $\chi^{(2)}$ waveguide. **b** The pump spectrum centered at 2.4 µm. **c** The measured OPG spectra from the $\chi^{(2)}$ waveguide with different pump peak powers. The generated LWIR idler wave has an octave-spanning spectrum covering 5.4 to 10.3 µm at −20 dB, which is in good agreement with the simulated result in Fig. 1a. The noise floor is marked by a gray dotted line. **d** The measured OPG output power shown in a linear scale as a function of pump pulse energy and average power. The measured pump threshold energy is ~616 pJ, corresponding to a peak power of 1.9 kW. In addition, When the pump energy is 6.5 nJ corresponding to an average power of 3.25 mW, a LWIR power of 0.72 mW is

detected from the $\chi^{(2)}$ waveguide, corresponding to a power efficiency of 22% and a quantum efficiency of 74%. Meanwhile, further increasing the pump energy, the output power tends to saturate. Particularly, when the pump pulse energy is larger than 9 nJ, the LWIR output power decreases, due to parametric back conversion. **e** The OPG output intensity in a logarithm scale at different pump energy. A saturated parametric gain of >58.6 dB/cm calculated from the measured maximum idler power (0.83 mW) and minimum idler power (1.1 nW) is observed at a pump energy of 7.9 nJ, which indicates that a decent parametric gain could be provided by the $\chi^{(2)}$ waveguide device. The inset shows the measured beam profile of LWIR OPG from the $\chi^{(2)}$ waveguide, which agrees well with the simulated field profile shown in Fig. 1b.

equations. Parametric seed with noise field represented by a complex Gaussian distribution with zero mean and a half-a-photon energy variance is adopted to mimic vacuum fluctuations[23]. The simulation results shown in Fig. 4d–f qualitatively agree well with the experimental results. We therefore suggest that spectral shaping of the demonstrated micro-waveguide-based parametric device could be realized by tuning the pump wavelength, which would broaden its applications in MIR spectroscopy and metrology. In addition, it is worthy to note that the repetition of simulations with and without including $\chi^{(3)}$ nonlinearity is conducted, revealing a nearly identical spectral evolution of the OPG process. Hence, it could be concluded that the generated idler wave is dominated by the $\chi^{(2)}$ nonlinear process. The detailed information about $\chi^{(3)}$ nonlinearity in the demonstrated parametric waveguide could be found in Supplementary Note 3. Meanwhile, a qualitative analysis of partial coherence characteristics of the OPG is also demonstrated in Supplementary Note 3, which needs further investigation in future.

## Discussion

Over the past decade, LWIR single-pass parametric sources have been demonstrated in bulk nonlinear crystals facilitated either by

birefringence or quasi-PM techniques. However, the quantum efficiency is usually low, limited by factors such as spatial/temporal walk-off, non-perfect beam overlap, large interaction area and short parametric coupling length associated with the Gaussian beam focusing geometry. Figure 5a overviews and compares reported quantum efficiencies of the state-of-art single-pass parametric conversions including OPA[18,24–28] (blue squares), OPG[29–31] (pink circles), DFG[32–38] (orange diamonds), and intra-pulse difference-frequency generation[3,39–45] (IPDFG) (green triangles) based on bulk nonlinear crystals in the LWIR range ($\lambda > 5$ µm), with that of the demonstrated $\chi^{(2)}$ micro-waveguide device (red star). Typically, OPAs/OPGs in bulk nonlinear crystals operate with microjoule or even higher pump energy. DFGs and IPDFGs pumped by mode-locked lasers at high-repetition (~MHz magnitude) allow lower pump pulse energy of nanojoules level, but the quantum conversion efficiency is limited to ~20% or lower. Owing to the tight pump-idler mode confinement and enhanced parametric interaction, a quantum efficiency of 74% at a small pump energy of ~6.5 nJ is obtained in the $\chi^{(2)}$ waveguide platform, which is enhanced by more than twofolds compared to the traditional single-pass parametric conversions in bulk crystals. In addition, another merit of the demonstrated $\chi^{(2)}$ waveguide device is reflected as a low threshold. As displayed in Fig. 5b, the measured

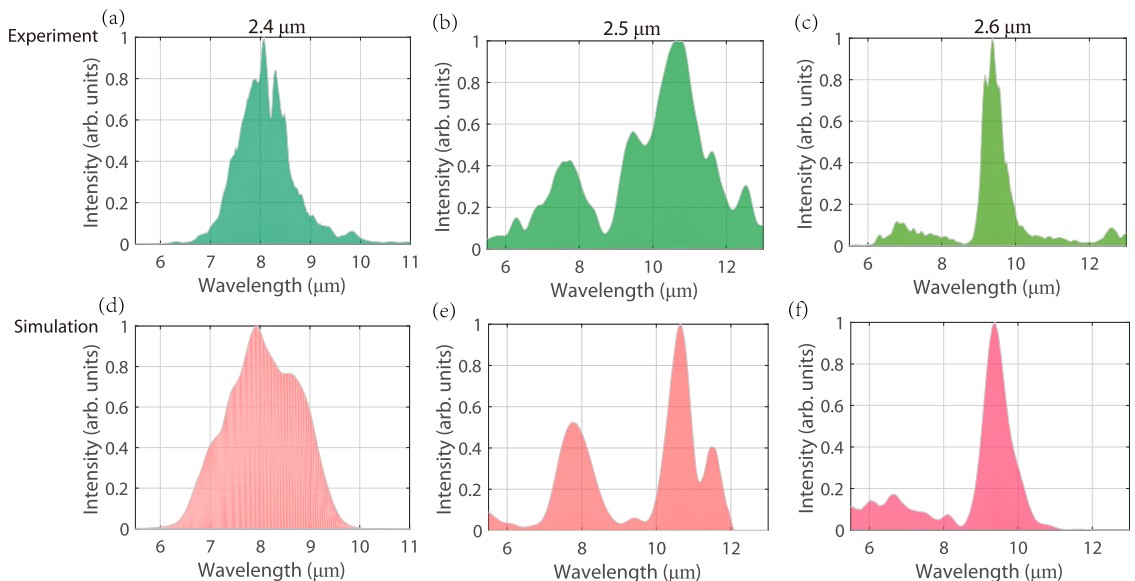

**Fig. 4 | The characterization of broadband spectral tuning and shaping from the χ⁽²⁾ micro-waveguide device with birefringence PM, when scanning the pump wavelength.** a–c and d–f are the measured and simulated optical parametric generation (OPG) spectra pumped at 2.4 μm, 2.5 μm and 2.6 μm, respectively. **a** A LWIR idler spectrum centered at 8 μm is measured pumped at 2.4 μm wavelength. **b** As the pump wavelength is increased to 2.5 μm, an octave-spanning idler covering 4.5 to 12 μm is generated, owing to a better PM condition at two wings of the spectrum. **c** Further increasing the pump wavelength to 2.6 μm, narrow-band idler spectrum centered at 9.3 μm is measured due to the reduced parametric bandwidth. **d–f** A semi-classical simulation of OPG is also conducted to study the spectral evolution in the χ⁽²⁾ waveguide with birefringence PM at different pump wavelengths, by solving the χ⁽²⁾-based coupled-wave equations. The measurement and simulation results qualitatively agree well with each other.

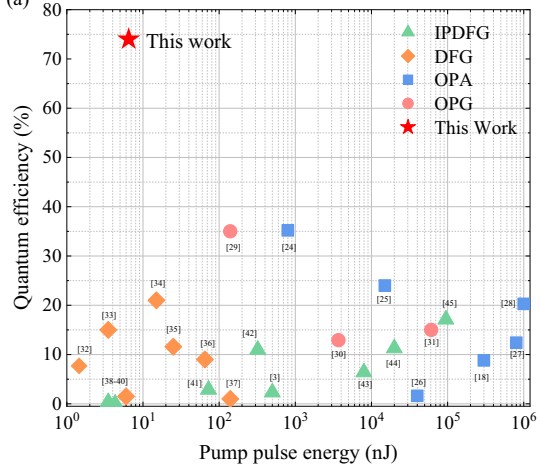
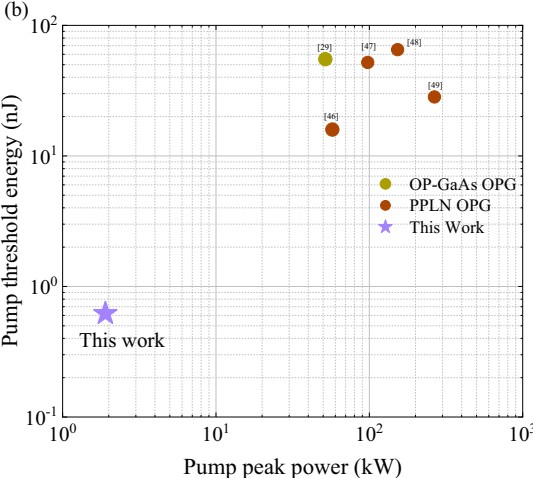

**Fig. 5 | Comparison of quantum efficiency and pump threshold of the demonstrated χ⁽²⁾ micro-waveguide based on birefringence PM with the state-of-art single-pass parametric conversions.** **a** Summary of quantum efficiencies and saturated pump pulse energy of single-pass parametric sources including representative OPA[18,24–28], OPG[29–31], DFG[32–38] and IPDFG[3,39–45] in the LWIR region (>5 μm) based on bulk nonlinear crystals, and the demonstrated OPG based on ZGP χ⁽²⁾ micro-waveguide platform. A quantum efficiency of 74% at a pump pulse energy of 6.5 nJ is demonstrated, which is enhanced by more than 2 folds compared to the state-of-art LWIR single-pass parametric conversions in bulk crystals. **b** Comparison of the threshold pulse energy and peak power of the demonstrated OPG in the χ⁽²⁾ waveguide based on birefringence PM and other MIR OPGs with quasi-PM, based on PPLN[46–49] and OP-GaAs crystals[29]. The measured threshold pump energy of ZGP χ⁽²⁾ waveguide is 616 pJ, corresponding to a peak power of 1.9 kW, which is reduced by more than 1-order of magnitude compared to the reported MIR OPGs in the literature.

threshold pulse energy and peak power from the demonstrated χ⁽²⁾ waveguide is compared with those of OPGs in χ⁽²⁾ bulk media including PPLN[46–49] and OP-GaAs[29] crystals. A low threshold pulse energy and peak power measured as 616 pJ and 1.9 kW are obtained, respectively, which is more than 1-order of magnitude lower than those of reported OPGs in literature. In addition, to further reveal the technical breakthrough of the demonstrated ZGP micro-waveguide, the performance comparison with reported PPLN nano-waveguide[12], and OP-GaAs waveguide[15] is also

executed (Supplementary Note 6). The OPG in the PPLN nano-waveguide[12] exhibits extraordinary threshold energy, but a spectral range of 1.7–2.7 μm, limited by the transparency window of the PPLN crystal, and a relatively low quantum conversion efficiency (22.2%). Based on OP-GaAs crystal, as an attractive MIR nonlinear optical material, it presents an OPG output with the spectrum extending to LWIR region, which is a significant breakthrough in the field of MIR integrated photonics; however, highly efficient single-pass parametric conversion

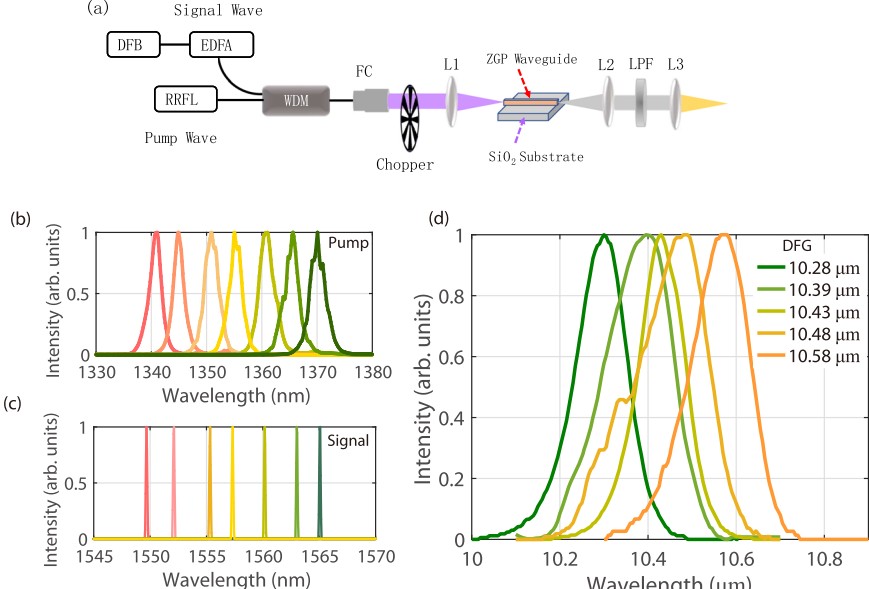

**Fig. 6 | Experimental setup of c. w. DFG based on another ZGP micro-waveguide and characterizations of spectra of pump, signal, and corresponding LWIR idler waves.** a The schematic of c. w. mid-infrared laser generation through DFG in ZGP waveguide. WDM wavelength division multiplexer, L lens, FC fiber collimator, LPF long-pass filter. The setup is composed of three parts, including a home-built c. w. tunable random Raman fiber laser (RRFL) at a wavelength around 1350 nm, which is used as DFG pump source, as depicted in Supplementary Note 4, a commercial erbium-doped fiber amplifier (EDFA) seeded by tunable

signal-frequency DFB laser (Conquer, KG-TLS-13-P-FA) emitting in the spectral range of 1527–1567 nm as DFG signal source, and another ZGP waveguide, cut at $\theta = 65.4°$ and $\varphi = 0°$. The RRFL and EDFA sources are combined by a fiber wavelength division multiplexer (WDM) as an all-fiber laser driver for DFG in the ZGP waveguide. **b** Typical pump spectra tuning from 1340 to 1370 nm. **c** Typical signal spectra tuning from 1550 to 1565 nm. **d** The measured spectra of generated LWIR lasing tunable from 10.28 to 10.58 μm through DFG in the ZGP waveguide.

has not been achieved. Notably, although ZGP micro-waveguide-based OPG has a relatively high pump threshold compared to that of PPLN nano-waveguide, benefited from waveguide-enhanced parametric interaction and inherently strong quadratic nonlinearity, a remarkable parametric quantum conversion efficiency together with the broadband and tunable LWIR spectrum across 5–12 μm is successfully and firstly realized. These overviews and comparisons of MIR single-pass parametric sources show the demonstrated quadratic nonlinear waveguide as a marked advancement of compact and efficient MIR laser sources.

It is worthy to mention that although the remarkable parametric conversion efficiency of 74% in the LWIR region is obtained in the demonstrated ZGP micro-waveguide platform, the required peak power to realize OPG is still in -kW level, which means only pulse pump scheme is feasible. To miniaturize the whole system in near future for practical applications, continuous-wave (c. w.) pump is highly desired. To demonstrate the prospect of ZGP micro-waveguide towards integrated photonics, a proof-of-concept experiment of c. w. DFG driven by an all-fiber laser source in another ZGP waveguide cut at $\theta = 65.4°$ and $\varphi = 0°$ is conducted. Figure 6a illustrates the experimental setup of c. w. DFG in ZGP Waveguide. The setup includes a home-built c. w. tunable random Raman fiber laser (RRFL) around the wavelength of 1350 nm serving as the DFG pump (the detailed laser configuration is depicted in Supplementary Note 4), and a commercial erbium-doped fiber amplifier (EDFA) seeded by a tunable signal-frequency DFB laser (Conquer, KG-TLS-13-P-FA) in the wavelength range of 1527–1567 nm as the DFG signal source. The RRFL and the EDFA output are combined by a fiber wavelength division multiplexer to form an all-fiber laser driver for DFG in the ZGP waveguide. Although the ZGP waveguide has a fixed PM angle, the tunable DFG radiation can be obtained by simultaneously tuning the wavelengths of pump and signal light to fulfil PM conditions. The typical tunable spectra of the pump and signal laser sources are presented in Fig. 6b, c, respectively. As illustrated in Fig. 6d, when the power of pump and signal light is set both as

250 mW, through DFG in ZGP waveguide, tunable LWIR lasing across 10.28–10.58 μm can be obtained, which indicates that c. w. pumped ZGP waveguide could be a promising platform for integrated tunable LWIR light generation. More advanced coupling methods such as using lensed fiber coupling can be adopted to further improve the compactness of the system. The results also imply that in future, by combining the tunable near-infrared (NIR) single-frequency laser diodes and the developed ZGP micro-waveguide, tunable single-frequency LWIR light can be realized in an integrated waveguide platform, and such an on-chip single-frequency LWIR light source could be a desirable tool for LWIR spectroscopy and free-space communication applications[50].

In summary, we, for the first time to the best of our knowledge, propose and demonstrate a $\chi^{(2)}$ nonlinear micro-waveguide platform based on the birefringence PM for simple and highly efficient LWIR generation. Tunable LWIR spectra in the wavelength range of 5–12 μm are achieved by scanning the pump wavelength. With the waveguide-enabled tight mode confinement and improved $\chi^{(2)}$ parametric interaction, a record LWIR single-pass quantum efficiency of 74% is demonstrated. High-gain (>58.6 dB) and low-threshold (616 pJ) LWIR OPG with an octave-spanning bandwidth is also realized in the $\chi^{(2)}$ waveguide. In addition to the pulse-pumped OPG, a proof-of-concept demonstration of more compact c. w. LWIR tunable DFG is performed in ZGP micro-waveguide pumped by an all-fiber laser source, manifesting the prospect of ZGP micro-waveguide as an integrated photonics platform to generate LWIR light for portable applications.

We anticipate that our work opens new possibilities to generate LWIR light sources and perform nonlinear frequency conversion-based applications in LWIR on an integrated photonics platform. First, the waveguide fabrication technique introduced in this work is simple, flexible and universal to various $\chi^{(2)}$ birefringence crystals including $AgGsS_2$, $LiGaS_2$ and $CdSiP_2$, for different parametric conversions and applications. For example, the $LiGaS_2$ micro-waveguide can enable LWIR parametric conversion pumped by commercially mature 1 μm fs

lasers. Second, the demonstrated record high LWIR single-pass parametric quantum efficiency in ZGP waveguide manifests that the $\chi^{(2)}$ nonlinear micro-waveguide platform can lead to a new generation of highly efficient and broadband tunable MIR laser sources, especially in the LWIR region. To improve the system compactness, more compact femtosecond lasers operating around 2.4 μm with decent output power can be adopted. Kerr-lens mode-locked Cr$^{2+}$: ZnS/ZnSe oscillators[51] and MIR fiber lasers based on soliton self-frequency shift[52] could output up to hundreds of kW peak power which is well above the pump threshold of the demonstrated $\chi^{(2)}$ waveguide-based single-pass parametric generator. It is also worth mentioning that the micrometer size of the $\chi^{(2)}$ waveguide is beneficial for realization of high efficiency light coupling with single-mode lensed fiber, making the developed $\chi^{(2)}$ waveguide platform fiber-compatible with high overall efficiency. We also expect more advanced explorations of MIR integrated nonlinear photonics such as on-chip IPDFG[53], MIR femtosecond lasers with ultra-high repetition rate, and $\chi^{(2)}$ frequency comb generation[54] could be pursued in the near future. In addition, the proof-of-concept demonstration of c. w. all-fiber laser pumped DFG in $\chi^{(2)}$ ZGP waveguide provides another feasible and reliable route for system miniaturization, and the combination of $\chi^{(2)}$ waveguide and NIR c. w. fiber lasers or laser diodes provides possibilities to realize on-chip nonlinear frequency converter, which could be a versatile platform for various promising applications in LWIR region such as up-conversion spectroscopy/imaging[55] and quantum photon-pair generation[56]. On the other hand, by further developing high-finesses $\chi^{(2)}$ waveguide or micro-resonator, c. w.-pumped on-chip optical parametric oscillator in LWIR could be pursued in future. We therefore believe that the $\chi^{(2)}$ micro-waveguide device with birefringence PM fabricated by the simple technique in this work, equipped with advanced pump sources would flourish the MIR integrated photonics researches and promote practical applications of MIR spectroscopy and metrology with compact systems.

## Methods

### The pump source: KTiOPO$_4$ (KTP)-based OPA

The pump source starts with a customized Yb-fiber laser (Yacto, YF-FL-50-100-IR) emitting 1030 nm pulses with a duration of 260 fs at 500 kHz repetition rate. A small fraction of pulse energy ~8 μJ is focused into a 15-mm-long YAG crystal generating a stable white light spanning from 1.1 to 1.9 μm in the near-infrared region. An 8-mm-long anti-reflection coated KTP crystal cut at $\theta$ = 44.5°, $\varphi$ = 0° for type II PM is used in the OPA. The idler wavelength is tunable from 2.4 to 2.8 μm by twisting the PM angle. The output power of the idler wave is measured up to 500 mW by using a 15 W pump. The pulse duration of idler wave is characterized as 320 fs with some uncompensated dispersion inherited from the OPA system, through a home-built interferometer autocorrelator.

### $\chi^{(2)}$ waveguide fabrication

A bulk ZGP crystal cut into a PM angle of 48.3° for type I PM is bonded to a fused silica substrate using ultraviolet curing optical adhesives (NOA61). The bonded ZGP crystal is subsequently lapped and polished by a plane precision ring polishing machine (Jian Su XB, LP10C/08C) to a thickness of ~40 μm. In the process of ultrafast laser direct writing, the 520 nm femtosecond laser beam (Spectra-Physics Spirit HE 1040-30-SHG) with a pulse duration of 300 fs at a repetition rate of 250 kHz is focused to the ZGP surface by an F-Theta lens with a focal spot size of 10 μm in diameter. A coaxial charge-coupled device imaging system is equipped to achieve an accurate focusing. The laser processing parameters are optimized with the laser fluence, scanning speed, scanning spacing, and processing cycles as 1.53 J/cm$^2$, 100 mm/s, 5 μm, and 2, respectively. A vacuum cleaner is turned on during the laser processing to remove the debris produced in the femtosecond laser fabrication. The $\chi^{(2)}$ wafer is cleaned using an ultrasonic cleaner after laser

processing in deionized water for 5 seconds. With above procedures and parameters, the waveguide side-wall roughness is measured to be less than 0.6 μm, and the waveguide facets are nearly identical to those of unprocessed ZGP films.

### Calibrations of the coupling and collecting efficiency

The coupling efficiency is measured by replacing the $\chi^{(2)}$ waveguide with a pinhole of 100 μm in diameter, and record the transmitted power ratio of the 2.4 μm laser beam. The coupling efficiency is measured as 15.3% which is several times larger than those of PPLN nano-waveguides, benefited from the multi-wavelength-scale dimensions in the fabricated $\chi^{(2)}$ waveguide device (More detailed information of coupling efficiency measurement is demonstrated in Supplementary Note 7). Notably, nonideal overlap of the TM$_{00}$ mode of waveguide and the mode profile at the focal plane of the lens induces the relatively high coupling loss that, however, could be significantly improved by using tapered edge couplers[57]. On the collecting side, the generated LWIR idler radiation is collected by a 1-inch ZnSe lens with a focal length of 25 mm, corresponding to a numerical aperture (NA) of 0.45, and an anti-reflection coating in the spectral range of 2–13 μm. Using the method demonstrated in refs. 12,13, the OPG average photon number could be expressed as $\langle n \rangle = a \sin h^2(2gl)$, where $a$ is the overall detection efficiency including the output-coupling loss and imperfect detection induced loss, $g$ indicates the parameter gain coefficient. Here, $g$ is in directly proportion to $\sqrt{\eta P}$, where $\eta$ represents the nonlinear interaction coefficient and $P$ is the pump power. When the parametric gain is larger than 10 dB, $\langle n \rangle$ could be approximated as $\langle n \rangle = a \exp(2gl)$. In addition, the measured average OPG power could be expressed as $P_{OPG} = \langle n \rangle h v f_{rep}$, from which $\langle n \rangle$ could be fitted and obtained, where $h$ is the Planck constant, $v$ represents the idler frequency and $f_{rep}$ indicates the repetition frequency. Hence, the parameter $a$ could be estimated, and the collection efficiency is obtained as 34%.

### Measurement equipment parameters

The employed spectral analyzer is a grating-scanning monochromator (Zolix Omni-λ500i) equipped with a lock-in amplifier (SRS, SR830) and a liquid nitrogen Mercury Cadmium Telluride (MCT) detector (Judson, DMCT16-De01). The monochromator resolution is 0.4 nm in the wavelength range of 2–16 μm. The typical responsibility is 900 V/W at 14 μm. The average power of generated MIR idler wave is detected using a MCT detector connecting a lock-in amplifier (SRS, SR830) and a power meter (Ophir, 3A, power range: 10 μW–3 W @ 0.19–19 μm). The typical detectable threshold and the measurement sensitivity of the MCT employed with the preamplifier (×100 magnification) is 0.3 nW @14 μm.

## Data availability

The simulation data generated in this study are provided in the Source data file. All other data used in this study are available from the corresponding author upon request. Source data are provided with this paper.

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

## Acknowledgements

This work was supported by National Natural Science Foundation of China (62075144, 12175157, 62005186, 62375189), Sichuan Outstanding Youth Science and Technology Talents (2022JDJQ0031), Engineering Featured Team Fund of Sichuan University (2020SCUNG105), and Shenzhen key Project for Technology Development (JSGG20191129105838333).

## Author contributions

H.L. conceived and designed the experiment. B.H. X.Y., X.L., and H.W. carried out the experiment of OPG and c. w. DFG measurement. H.L., Y.L., H.W., C.G. and Q.J.W. designed ZGP waveguide. B.H., J.W., W.L. and Huan Yang fabricated ZGP waveguide. B.H., X.Y., S.L., Hang Yang, and Z.L. conducted the theoretical simulations. X.Y., K.T., L.H., and W.W. built the MIR OPA. H.L., B.H., X.Y., J.W., Huan Yang, and H.W. wrote the manuscript. All authors discussed the results and contributed to the manuscript.

## Competing interests

The authors declare no competing interests.
