## [Peer Review File · Nature Communications]

Highly efficient octave-spanning long-wavelength infrared generation with a 74% quantum efficiency in a $\chi(2)$ waveguideREVIEWER COMMENTS

Reviewer #1 (Remarks to the Author):

In the manuscript entitled “On-chip highly efficient octave-spanning long wavelength mid-infrared optical parametric generation with a 74% quantum efficiency”, the authors present a ZGP-based integrated waveguide for an octave-spanning spectrum covering 5-11 μm , with birefringence phase matching. The quantum conversion efficiency and threshold energy are well improved compared with other methods. This topic is important and this work reports some interesting results. Also the manuscript is written clearly, containing many technical information to reproduce the results. However, I find overall that, this work is very incremental and presents poor novelty. The employed methods are well known by now (ultrafast laser direct writing, broadband phase matching, etc.) and the theoretical simulation is simplistic.

In addition, I have a list of additional questions and remarks, hopefully which could help to make this work more unique and comprehensive.

1. The authors point out some non-oxide nonlinear crystals such as ZGP, AgGsS₂, GaSe, and CdSiP₂, but only ZGP is used in this paper. How about other $\chi(2)$ crystals using the same fabrication technique? What are their shortcomings and limitations? More descriptions concerning this issue may be meaningful, to attract broad interest.
2. In Fig. 3(d), an obvious drop of output power appears around 9 nJ pump pulse energy, why and can it be predicted in theoretical simulation?
3. How about the calculated coupling coefficient? Taking the pump laser property into consideration (at least spot size and mode field of the output beam), to which degree can it be improved by further special design (like grating or tapered structures)?
4. Detailed information about the measurement equipment used in experiment is necessary. What is the performance for the employed spectral analyzer? The resolution, the sensitivity and detectable threshold?
5. All information about the “signal” against the “idler” is lacked. Actually it is import to investigate such characteristics since it is more available in experiment, and probably it can help to dig out some new physical mechanism behind.
6. How about the coherence across the full spectrum? If there exists a way to measure the output “pulse” (if so) duration? Does the $\chi(3)$ process such as SPM and FWM happened, due

to this waveguide structure?

7. Despite the threshold energy is found to be $<1\text{nJ}$, yet the required peak power is still in $\sim\text{kW}$ level, which means only the pulse pump scheme is feasible. To minimize the whole system in near future for practical applications, the c.w. pump is highly desired. Is it accessible with all possible efforts for now, such as the optimization of waveguide cross section or trying the OPO method through a microring?

In conclusion, although the authors have done a good job, given the above and doubtful novelty of the results, I would not recommend the manuscript for publication in Nature Communications. However, I do encourage the authors to rapid publish the current work in other related journals (e.g., those for optical materials and fab techniques).

Reviewer #2 (Remarks to the Author):

In this paper the authors report on the generation of broadband long wavelength MIR laser by using a ZGP waveguide fabricated by femtosecond laser writing to enhance second-order nonlinear effect. The performance, especially for conversion efficiency, of this waveguide for optical parametric generation is impressive. However, I cannot recommend the publication of this version of manuscript due to the concerns listed as below.

1)Although the results and the technical manifestation to be of high quality, I find the conceptual novelty to be somewhat lacking. Both physics and tech used in this manuscript including ZGP material, tight light confinement, large mode overlap, wafer bonding and femtosecond laser writing, are well-known in the literature. Less knowledge can be learned from the phenomenon and experimental description. I strongly recommend the authors to seriously summarize the novelty of this work, and focus on it, such as ZGP waveguide fabrication tech which is the key of high efficiency OPG.

2)The authors claim “integrated nonlinear photonics”, “integrated waveguide”, “on-chip”, many times in the contents of manuscript. But, in my opinion, the experimental medium is only a planar waveguide, more like a fiber but weakly related to integrated photonics. I do not see in this current realization any path to allow an integrated system, utilizing the large table-top pumping, coupling structure and filters. The authors should provide a detail discussion on the prospects of miniaturization and the performance of such envisioned system in line 214-223.

3) Why choosing 10mm length? What's the limitation?

4) In line 97, the authors claim the simulation loss of the waveguide is less than 0.25dB/cm, how about the real fabricated devices. I suggest giving a detail characterization of samples. In line 125, the roughness of the waveguide is 10um, which is compatible with processing wavelength, it's really not good for a low loss waveguide. If the roughness is 10um, the mode index changing rapidly, along waveguide. How about the influences for phase matching and other effects.

5) The experimental details and results need to be claimed clearly. Line 166, how can get the gain of waveguide is 60dB/cm? Are data in fig

Reviewer #3 (Remarks to the Author):

Hu et al. present experimental results on OPG using critical phase-matching in a 10-mm long, large-area ZGP waveguide pumped with 320-fs pulses at 2.4 um and producing an octave-spanning idler around 8 um. The use of critical phase matching on a waveguide structure is achieved by misaligning the waveguide with respect to the crystal axes while using a TE mode for the pump, and TM modes for the signal/idler. The device development and the presented results are significant, especially with respect to the spectral coverage, threshold, and efficiency. However, the paper needs substantial attention in terms of the accuracy of several claims and statements as well as better positioning of the work with respect to state of the art. Here are some detailed comments:

- While the authors cite other integrated nonlinear photonic platforms, such as recent demonstration of OPG in PPPLN [11,12] and OP-GaAs [14], they don't do direct comparisons to the numbers presented in those works. The threshold energies in these other integrated platforms are lower by several orders of magnitude, and while not explicitly claimed in the papers, the quantum efficiencies should also be comparable (if not) higher. Thus, the statement about "a new record in conversion efficiency in MIR" needs to be substantiated with a more comprehensive comparison. The current comparison seems to only include bulk crystals.

- In the presented comparisons, the authors state that they are comparing to "state-of-the-art parametric processes." However, they are only including single-pass parametric processes. Using bulk OPOs, similarly high mid-IR conversion efficiencies and low thresholds

have been achieved. This should be either included in the comparison or the statement needs to be revised.

- The authors mention in their abstract that X(3) has an inherently low conversion efficiency, and they expand on this in their introduction. While the strength of the nonlinearity is known to be low for X(3) processes, the conversion efficiency, however, is not necessarily low. Examples of X(3) processes with higher efficiencies include recent demonstrations of Kerr combs using pulsed pumps and dark soliton generation. Furthermore, the supercontinuum processes they cite based on X(3) interactions in integrated platforms (such as ref. [8]) exhibit lower thresholds for the onset of supercontinuum generation and don't seem to face an issue in terms of conversion efficiency. Such comparisons with X(3) can be more accurate.

- It seems, from Figure 2a, that the optical axis of the crystal is in-plane. Therefore, the TM modes should be “ordinary” since they are polarized orthogonal to the optical axis. The TE pump is then extraordinary. The authors make the opposite statement in line 91. Is this a typo? Or is Fig 2a not clear regarding the orientation of the optical axis?

- The modes of anisotropic waveguides can be considerably lossy when propagation happens in a direction not aligned with the crystal axes [J. Opt. Soc. Am. A 10, 246-261 (1993)]. The results of this work suggest this is not the case here. Could the authors explain why?

- The phase-matching mechanism used can be interpreted as modal phase matching. This has been demonstrated before in other platforms, but it has been usually weak. The results of this work are enabled by the large nondiagonal tensor (d_{36}) of ZGP. The paper can benefit from such a generalization and comparison with other materials.

- The authors claim “dispersion engineering” in line 104, which is not accurate as the waveguide geometry doesn't seem to affect the overall dispersion. The authors even explicitly mention that in lines 106-108 “... Owing to the multi-wavelength-scale geometry, the effect of waveguide dispersion is weak, and thus the ZGP waveguide features a nearly identical GVD profile with that of the bulk material...”. The angle they choose cannot be exploited as an independent degree of freedom for dispersion engineering since it's already exhausted by the phase-matching condition. This is a major point of confusion.

Other less major comments:

- The side-wall roughness mentioned on line 125 ... should it be less than 10 nm as opposed to 10 μm ?
- The sentence on the drawbacks of QCLs needs structural revisions.
- "long-wavelength mid-infrared" is probably better changed to mid-to-long-wavelength infrared or something like that. Typically, MWIR and LWIR are distinct, and the source seems to span both.
- Measurement of the input coupling using a pinhole approximating the size of their mode seems to neglect the effects of the AR coatings, mode mismatch, scattering, etc. First, this method needs to be justified. Second, how the accuracy of such a measurement can affect the claimed numbers should be discussed in more depth.

We thank the reviewers for the constructive comments of our work and useful suggestions for improving the manuscript. Please find below the response to all the comments.

Reviewer #1

In the manuscript entitled “on-chip highly efficient octave-spanning long wavelength mid-infrared optical parametric generation with a 74% quantum efficiency”, the authors present a ZGP-based integrated waveguide for an octave-spanning spectrum covering 5-11 μm , with birefringence phase matching. The quantum conversion efficiency and threshold energy are well improved compared with other methods. This topic is important and this work reports some interesting results. Also the manuscript is written clearly, containing many technical information to reproduce the results. However, I find overall that, this work is very incremental and presents poor novelty. The employed methods are well known by now (ultrafast laser direct writing, broadband phase matching, etc.) and the theoretical simulation is simplistic.

Response: We thank the reviewer for the careful reading, nice summary of the manuscript, and valuable comments. We agree well with reviewer that the methods employed in our work including ultrafast laser direct writing and broadband phase matching are known techniques. However, we would like to stress that in this work employing the existing waveguide fabrication techniques including ultrafast laser direct writing, **we for the first time demonstrate the birefringence crystals-based micro-waveguide for the highly efficient LWIR generation, which generates a quantum conversion efficiency of 74% as a new record in LWIR single-pass parametric processes. The threshold energy is measured as ~ 616 pJ, reduced by more than 1-order of magnitude as compared to those of MIR optical parametric generations in bulk media.** Owing to the inherently efficient quadratic nonlinear response, waveguide structure improved spatial confinement and elongated interaction length, and good coupling efficiency derived from multi-micrometer dimension design in the $\chi^{(2)}$ waveguide with birefringence phase matching, **the extraordinary parametric performance is achieved, which represents a technical breakthrough** in the field of mid-infrared light generation.

In addition, **c. w. all-fiber laser pumped DFG in $\chi^{(2)}$ ZGP waveguide** is demonstrated as the proof-of-concept. Tunable LWIR lasing across 10.28-10.58 μm is obtained, which indicates that c. w. pumped ZGP waveguide could be a promising platform for integrated tunable LWIR light generation. More advanced coupling methods such as lensed fiber coupling can be adopted to further improve the compactness of the system. The combination of $\chi^{(2)}$ waveguide and NIR c. w. fiber lasers or laser diodes provides possibilities to realize on-chip nonlinear frequency converter, which could be a versatile platform for various promising applications in LWIR region such as up-conversion spectroscopy/imaging and quantum photon-pair generation.

Thirdly, we have shown that using ultrafast laser direct writing, the $\chi^{(2)}$ birefringence crystals-based micro-waveguide can be successfully fabricated. Such fabrication method could be extended to various $\chi^{(2)}$ birefringence crystals including ZGP, AgGsS₂, LiGaS₂ and CdSiP₂, **and therefore the LWIR light generation and applications based on $\chi^{(2)}$ birefringence crystals can now be performed on the waveguide platforms.**

We therefore believe that although ultrafast laser direct writing and broadband phase matching are known techniques, the demonstrated $\chi^{(2)}$ micro-waveguide device with birefringence phase matching fabricated by the simple technique would flourish the MIR integrated photonics researches and promote practical applications of MIR spectroscopy and metrology with very compact systems.

We have extracted and highlight this point in the revised version of the manuscript:

In abstract:

“The realization of compact and efficient broadband mid-infrared (MIR) lasers has enormous impacts in promoting MIR spectroscopy for various important applications. A number of well-designed waveguide platforms have been demonstrated for MIR supercontinuum and frequency comb generations based on cubic nonlinearities, but unfortunately third-order nonlinear response is inherently weak. Here, we propose and demonstrate for the first time a $\chi^{(2)}$ micrometer waveguide platform based on birefringence phase matching for long-wavelength infrared (LWIR) laser generation with a high quantum efficiency. In a ZnGeP₂-based waveguide platform, an octave-spanning spectrum covering 5 - 11 μm is generated through optical parametric generation (OPG). A quantum conversion efficiency of 74% as a new record in LWIR single-pass parametric processes is achieved. The threshold energy is measured as ~ 616 pJ, reduced by more than 1-order of magnitude as compared to those of MIR OPGs in bulk media. Our prototype micro-waveguide platform could be extended to other $\chi^{(2)}$ birefringence crystals and trigger new frontiers of MIR integrated nonlinear photonics.”

In Introduction:

“Up to now, nonlinear frequency conversions have become the main route for generating coherent ultra-broadband MIR radiations, which however generally consist of large and complicated laser apparatuses and exhibit relatively low conversion efficiency^{2,3}.”

“However, $\chi^{(3)}$ nonlinear response is inherently weak. Hence, dedicated dispersion engineering, moderate-to-small effective mode area, and resonators with high quality factors are usually required to achieve a reasonable conversion efficiency^{4-6, 9-11}. On the other hand, more efficient quadratic nonlinearity ($\chi^{(2)}$)-based waveguide devices for parametric conversions such as optical parametric generation/amplification (OPG/OPA), and difference-frequency generation (DFG) are expected to be a promising alternative approach for simple and efficient generation of ultra-broadband MIR lasers.”

“In this work, we, for the first time to the best of our knowledge, propose and demonstrate the $\chi^{(2)}$ parametric micro-waveguide platform with birefringence PM for highly efficient single-pass LWIR generation, taking the advantages of high birefringence $\chi^{(2)}$ in non-oxide nonlinear crystals, such as ZnGeP₂ (ZGP), AgGsS₂, GaSe and CdSiP₂ which are attractive for broadband LWIR generation¹⁶⁻¹⁹.”

“The demonstrated prototype micro-waveguide platforms could be extended to other $\chi^{(2)}$ birefringence crystals for highly efficient and broadband tunable MIR laser generation.”

In Discussion:

“It is worthy to mention that although the remarkable parametric conversion efficiency of 74% in the LWIR region is obtained in the demonstrated ZGP micro-waveguide platform, the required peak power to realize OPG is still in $\sim\text{kW}$ level, which means only pulse pump scheme is feasible.

To miniaturize the whole system in near future for practical applications, continuous-wave (c. w.) pump is highly desired. To demonstrate the prospect of ZGP micro-waveguide towards integrated photonics, a proof-of-concept experiment of c. w. DFG driven by an all-fiber laser source in another ZGP waveguide cut at $\theta = 65.4^\circ$ and $\varphi = 0^\circ$ is conducted. Fig. 6(a) illustrates the experimental setup of c. w. DFG in ZGP Waveguide. The setup includes a home-built c. w. tunable random Raman fiber laser (RRFL) around the wavelength of 1350 nm serving as the DFG pump (the detailed laser configuration is depicted in Supplementary Note 4), and a commercial erbium-doped fiber amplifier (EDFA) seeded by a tunable signal-frequency DFB laser (Conquer, KG-TLS-13-P-FA) in the wavelength range of 1527 - 1567 nm as the DFG signal source. The RRFL and the EDFA output are combined by a fiber wavelength division multiplexer to form an all-fiber laser driver for DFG in the ZGP waveguide. Although the ZGP waveguide has a fixed PM angle, the tunable DFG radiation can be obtained by simultaneously tuning the wavelengths of pump and signal light to fulfil PM conditions. The typical tunable spectra of the pump and signal laser sources are presented in Fig. 6(b) and (c), respectively. As illustrated in Fig. 6(d), when the power of pump and signal light is set both as 250 mW, through DFG in ZGP waveguide, tunable LWIR lasing across 10.28-10.58 μm can be obtained, which indicates that c. w. pumped ZGP waveguide could be a promising platform for integrated tunable LWIR light generation. More advanced coupling methods such as using lens fiber coupling can be adopted to further improve the compactness of the system. The results also imply that in future, by combining the tunable near-infrared (NIR) single-frequency laser diodes and the developed ZGP micro-waveguide, tunable single-frequency LWIR light can be realized in an integrated waveguide platform, and such an on-chip single-frequency LWIR light source could be a desirable tool for LWIR spectroscopy and free-space communication applications⁵⁰.”

Fig. 6| Experimental setup of c. w. DFG based on another ZGP micro-waveguide and characterizations of spectra of pump, signal, and corresponding LWIR idler waves. (a) The schematic of c. w. mid-infrared laser generation through DFG in ZGP waveguide. WDM, wavelength division multiplexer; L, lens; FC, fiber collimator; LPF, long-pass filter. The setup is composed of three parts, including a home-built c. w. tunable random Raman fiber laser (RRFL) at a wavelength around 1350 nm, which is used as DFG pump source, as depicted in Supplementary Note 4, a commercial erbium-doped fiber amplifier (EDFA) seeded by

tunable signal-frequency DFB laser (Conquer, KG-TLS-13-P-FA) emitting in the spectral range of 1527-1567 nm as DFG signal source, and another ZGP waveguide, cut at $\theta = 65.4^\circ$ and $\varphi = 0^\circ$. The RRFL and EDFA sources are combined by a fiber wavelength division multiplexer (WDM) as an all-fiber laser driver for DFG in the ZGP waveguide. (b) Typical pump spectra tuning from 1340 nm to 1370 nm. (c) Typical signal spectra tuning from 1550 nm to 1565 nm. (d) The measured spectra of generated LWIR lasing tunable from 10.28 μm to 10.58 μm through DFG in the ZGP waveguide.

In Conclusion:

“We anticipate that our work opens new possibilities to generate LWIR light sources and perform nonlinear frequency conversion-based applications in LWIR on an integrated photonics platform. First, the waveguide fabrication technique introduced in this work is simple, flexible and universal to various of $\chi^{(2)}$ birefringence crystals including AgGaS₂, LiGaS₂ and CdSiP₂, for different parametric conversions and applications. For example, the LiGaS₂ micro-waveguide can enable LWIR parametric conversion pumped by commercially mature 1 μm fs lasers. Second, the demonstrated record high LWIR single-pass parametric quantum efficiency in ZGP waveguide manifests that the $\chi^{(2)}$ nonlinear micro-waveguide platform can lead to a new generation of highly efficient and broadband tunable MIR laser sources, especially in the LWIR region.”

In addition, I have a list of additional questions and remarks, hopefully which could help to make this work more unique and comprehensive.”

1. The authors point out some non-oxide nonlinear crystals such as ZGP, AgGaS₂, GaSe, and CdSiP₂, but only ZGP is used in this paper. How about other $\chi^{(2)}$ crystals using the same fabrication technique? What are their shortcomings and limitations? More descriptions concerning this issue may be meaningful, to attract broad interest.

Response: We thank the reviewer for the questions and good suggestions. Ultrafast laser direct writing (ULDW) technique has been employed for almost arbitrary structures fabrication with diverse materials compatibility in integrated photonics [R1]. In principle, AgGaS₂, GaSe, and CdSiP₂ and other nonlinear crystals could also be effectively processed by the ULDW technique as birefringence $\chi^{(2)}$ waveguide platforms. In our current work, ZGP is used with reasons summarized as following aspects. Firstly, ZGP has an excellent quadratic nonlinear coefficient (75 pm/V) which is higher than those of AgGaS₂ and GaSe, only lower than that of CdSiP₂ (84.5 pm/V); however, the transparent window of ZGP (0.74~12 μm) is broader than that of CdSiP₂ (0.5~9 μm). Second, high quality ZGP crystals cut into designed phase-matching angles are easy to be obtained with mature growth technologies. Unlike ZGP, GaSe could be cleaved only along the (001) plane (z-cut, $\theta=0^\circ$), which restricts its usefulness in birefringence $\chi^{(2)}$ waveguide, although GaSe has a broader transparent window. In addition, ZGP has a good thermal conductivity (35 W/mK) which eliminates any thermal related variations of the fabricated $\chi^{(2)}$ waveguide, in case there is any absorption. (In principle there is no obvious absorption loss from the ZGP waveguide.). With above considerations, we choose ZGP as a typical example of birefringence $\chi^{(2)}$ waveguide.

Besides AgGaS₂, GaSe, and CdSiP₂, nonlinear crystals with large bandgap energy, such as LiGaS₂ may also be fabricated through the ULDW technique into a $\chi^{(2)}$ waveguide, which could be pumped at ~ 1 μm wavelength. This would broaden the usefulness and impact of demonstrated birefringence $\chi^{(2)}$ waveguide.

Table R1 summarizes and compares the optical properties of some typical long-wavelength IR nonlinear crystals.

Table R1. Comparisons of typical long-wavelength IR crystals

Crystals	Nonlinear coefficient (pm/V)	Transparent range (μm)	Thermal conductivity (W/mK)	Bandgap (eV)	Reference
ZGP	75	0.74-12	35	2.1	R2
AgGaS ₂	12	0.47-13	1.4	2.7	R3
GaSe	57	0.65-18	2.0	2.1	R4
CdSiP ₂	84.5	0.5-9	13.6	2.45	R5
LiGaS ₂	5.8	0.32-11.6	5.1	4.15	R6, R7

More descriptions and discussions about the choice of nonlinear crystal have been added into the revised manuscript and Supplementary Information.

Please see the changes in Line 85 - 88 in the revised manuscript: “The ZGP crystal is chosen as the material platform for its high $\chi^{(2)}$ nonlinearity ($d_{36} \sim 75$ pm/V), mature growth technique, and broad MIR transparent range ($\sim 0.73 - 12$ μm)”.

Please see the changes in Supplementary Note 1, including Table R1.

[R1]. Liu, X. et al. Dry-etching-assisted femtosecond laser machining. Laser Photonics Rev. 11, 1600115 (2017).

[R2]. Sanchez, D. et al. 7 μm, ultrafast, sub-millijoule-level mid-infrared optical parametric chirped pulse amplifier pumped at 2 μm. Optica 3, 147-150 (2017).

[R3]. Migal, E. A. et al. Highly efficient optical parametric amplifier tunable from near-to mid-IR for driving extreme nonlinear optics in solids. Opt. Lett. 42, 5218-5221 (2017).

[R4]. Liu, K. et al. Multimicrojoule GaSe-based midinfrared optical parametric amplifier with an ultrabroad idler spectrum covering 4.2-16 μm. Opt. Lett. 44, 1003-1006 (2019).

[R5]. Lesko, D. M. B. et al. A six-octave optical frequency comb from a scalable few-cycle erbium fibre laser. Nat. Photonics 15, 281-286 (2021).

[R6]. Nikogosyan, D.N. Nonlinear optical crystals: a complete survey, 1st ed. New York, NY: Springer, 2005.

[R7]. Li, W. et al. Theoretical Study on the Intrinsic Source of the Large Thermal Conductivity of Li-Based Chalcogenide Nonlinear Optical Crystals: From AgGaS₂ to LiGaS₂.

2. In Fig. 3(d), an obvious drop of output power appears around 9 nJ pump pulse energy, why and can it be predicted in theoretical simulation?

Response: We thank the reviewer for the question. The drop of output power around 9 nJ pump pulse energy is due to the parametric back conversion, when signal and idler waves accumulate significant amount of energy relative to the pump wave, and the parametric coupling is saturated. A numerical simulation by solving the $\chi^{(2)}$ -based coupled-wave equations is executed, which reproduces the decline of output power at a pump power of 9 nJ, as shown in Fig. R1.

Fig. R1. The simulated output power from the ZGP $\chi^{(2)}$ waveguide as a function of the pump pulse energy, by solving the $\chi^{(2)}$ -based coupled-wave equations. The drop of output power around 9 nJ pump pulse energy is reproduced

3. How about the calculated coupling efficient? Taking the pump laser property into consideration (at least spot size and mode field of the output beam), to which degree can it be improved by further special design (like grating or tapered structures)?

Response: We thank the reviewer for the comments. The difference between the mode field diameters of the pump beam at the focal plane and the waveguide size influences the coupling efficiency significantly. In our work, the waveguide width is designed to be $\sim 40 \mu\text{m}$. The $2.4 \mu\text{m}$ pump beam is focused into the waveguide by using a CaF_2 lens with a focal length of 40 mm. A focused pump beam with a diameter of $34 \mu\text{m}$ at focus is calculated by using an input beam diameter of 5 mm and M^2 value of 1.4, which is comparable with that of waveguide dimension, indicating a predictable good pump coupling efficiency. In our work, considering the alignment tolerance, adopting reflective objective lens with high-numerical aperture to focus the pump beam into a smaller beam size could be a promising method to improve the coupling efficiency in a further step. Notably, as the reviewer mentioned, in future, adopting the tapered edge coupler structures with good AR coating to expand the waveguide modes at the input could also largely improve the coupling efficiency to $>90\%$ [R8]. In addition, a typical fundamental effective mode area of the $2.4 \mu\text{m}$ pump wave is calculated as $93 \mu\text{m}^2$ by COMSOL Multiphysics simulation tool. Thus, by using taper structures, the amount of pump power coupling into the fundamental mode of the waveguide could also be enhanced. More discussion on the methods for enhancing the coupling efficiency has been added into the revised manuscript.

Please see the changes in Line 328 - 331 in the revised manuscript: “Notably, nonideal overlap of the TM_{00} mode of waveguide and the mode profile at the focal plane of the lens induces the relatively high coupling loss which, however, could be significantly improved by using tapered edge couplers.”

The added new reference in the revised manuscript:

In Line 461 - 462: “57. Hu, C, et al. High-efficient coupler for thin-film lithium niobate waveguide devices. *Opt. Express* 29, 462 5397-5406 (2021).”

[R8]. Hu, C, et al. High-efficient coupler for thin-film lithium niobate waveguide devices. *Opt. Express* 29, 5397-5406 (2021).

4. Detailed information about the measurement equipment used in experiment is necessary. What is the performance for the employed spectral analyzer? The resolution, the sensitivity, and detectable threshold?

Response: We thank the reviewer for the comments and questions. Detailed information about the measurement equipment used in our study is added in the method part of the revised manuscript. The employed spectral analyzer is a grating-scanning monochromator (Zolix Omni- λ 500i) equipped with a lock-in amplifier (SRS, SR830) and a liquid nitrogen Mercury Cadmium Telluride (MCT) detector (Judson, DMCT16-De01). The monochromator grating resolution is 0.4 nm in the wavelength range of 2-16 μm . The typical responsibility is 900 V/W at 14 μm . The average power of generated MIR idler wave is detected using a MCT detector connecting with a lock-in amplifier (SRS, SR830) and a power meter (Ophir, 3A, power range: 10 μW -3W @ 0.19-19 μm). The typical detectable threshold and the measurement sensitivity of the MCT employed with the preamplifier (100 \times magnification) is 0.3 nW @14 μm .

The detailed information about the measurement equipment used in our study has been added in the revised manuscript. Please see the changes of the method part of the revised manuscript.

“Measurement equipment parameters. The employed spectral analyzer is a grating-scanning monochromator (Zolix Omni- λ 500i) equipped with a lock-in amplifier (SRS, SR830) and a liquid nitrogen Mercury Cadmium Telluride (MCT) detector (Judson, DMCT16-De01). The monochromator grating resolution is 0.4 nm in the wavelength range of 2-16 μm . The typical responsibility is 900 V/W at 14 μm . The average power of generated MIR idler wave is detected using a MCT detector connecting with a lock-in amplifier (SRS, SR830) and a power meter (Ophir, 3A, power range: 10 μW -3W @ 0.19-19 μm). The typical detectable threshold and the measurement sensitivity of the MCT employed with the preamplifier (100 \times magnification) is 0.3 nW @14 μm .”

5. All information about the “signal” against “idler” is lacked. Actually, it is important to investigate such characteristics since it is more available in experiment, and properly it can help to dig out some new physical mechanism behind.

Response: We thank the reviewer for the valuable suggestion. We are sorry that only idler information was shown in the original manuscript as we focused more on the long-wavelength infrared region. As shown in Fig. R2(a), the corresponding signal spectrum is measured and plotted together with the idler spectrum. The signal spectrum has a central wavelength located at \sim 3.45 μm , as predicted by the phase-matching condition. The output power of the signal wave as a function of the pump power is measured and displayed in Fig. R2(b). The maximum unsaturated signal power of 1.62 mW is obtained at a pump power of 3.25 mW, indicating a power conversion efficiency of \sim 50% and a quantum efficiency of 72% which is very similar to that of the idler efficiency

measurement. Meanwhile, in the low gain region, the signal power is comparable with that of idler power, revealing an inherent characteristic of OPG seeded by quantum noises [R9]. The signal information and corresponding discussions have been added in the revised manuscript and Supplement.

Fig. R2. The measured signal and idler spectra and power from the ZGP $\chi^{(2)}$ birefringence waveguide. (a) The spectra of signal and idler waves. (b) The output power of signal and idler waves as a function of pump power.

Please see the changes in Line 175 - 176 in the revised manuscript: “The output power of the generated MIR idler from the ZGP waveguide is plotted in Fig. 3(d), as a function of pump pulse energy and average power. (The corresponding signal information could be found in the Supplementary Information.)”

In the revised Supplementary Information:

The signal information has been added into the revised Supplementary Information. Please see the changes in Supplementary Note 2.

[R9]. Louisell, W. H. & Yariv, A. Quantum Fluctuations and Noise in Parametric Processes. I. Phys. Rev. 124, 1646-1654 (1961).

- How about the coherence across the full spectrum? If there exists a way to measure the output “pulse” (if so) duration? Does the $\chi^{(3)}$ process such as SPM and FWM happened, due to this waveguide structure?

Response: We thank the reviewer for the questions. The effect of $\chi^{(3)}$ processes such as SPM and FWM are investigated, as presented in Fig. R3. The 2.4 μm pump pulse with a pulse energy of 6 nJ and a pulse width of 320 fs is focused by a CaF₂ lens with a focal length of 40 mm into the $\chi^{(2)}$ ZGP waveguide. Incident and transmitted spectra are compared in Fig. R3(a). No obvious spectral broadening or side bands are observed, which indicates that $\chi^{(3)}$ effects such as SPM or FWM are negligible as the pump pulse propagating along the $\chi^{(2)}$ ZGP waveguide. In addition, a simulation of OPG in the $\chi^{(2)}$ ZGP waveguide is conducted based on coupled-wave equations with random noises as the input signal [R10-R12]. The simulated idler spectra with and without including the $\chi^{(3)}$

nonlinearity are compared, as shown in Fig. R3(b). Nearly identical spectra are simulated which means in the $\chi^{(2)}$ ZGP waveguide, $\chi^{(3)}$ effects do not make detectable contribution to the parametric process.

As for the coherence of the output from $\chi^{(2)}$ ZGP waveguide, the output of OPG is partially coherent, and output with picosecond or femtosecond pulses have been demonstrated in OPG [R13, R14]. We have a home-built second-harmonic generation-based interferometer autocorrelator (IAC) in our lab, however, limited by the MIR output power and pulse energy (0.8 mW, 1.6 nJ) from the $\chi^{(2)}$ ZGP waveguide, the temporal profile of the OPG radiation from the $\chi^{(2)}$ waveguide is difficult to be measured. Alternatively, for characterizing the coherence of OPG output, we reproduce the OPG experiment in a 10-mm-long bulk ZGP crystal (DPT, YS-ZGP cut at $\theta = 48.4^\circ$, $\varphi = 0^\circ$) with the same experimental condition. Similar idler spectra are produced from the bulk ZGP crystal and the ZGP $\chi^{(2)}$ waveguide. The home-built IAC is used to measure the temporal profile of the generated idler pulse. As presented in Fig. R4 (a), an interferometric trace with a 1:8 ratio between the background and the maximum of the IAC signal proves the reliability of the measurement result. The electric field is reconstructed through a genetic algorithm based on the “evolutionary phase retrieval from interferometric autocorrelation (EPRIAC)” algorithm [R15]. In the Taylor expansion of the reconstructed spectral phase, dispersion terms up to seventh order are considered. The retrieved IAC trace exhibiting a good match with the measured one and the reconstructed temporal profiles are shown in Fig. R4(a, b), respectively. A pulse width of ~ 270 fs is measured, which is much longer than the transform-limited pulse width (~ 60 fs) and close to the duration of the pump pulse (320 fs). Moreover, pulse splitting in some extent is revealed. The measurement thus verifies that the output of OPG is partially coherent, which is also accord with the previous work [R10-R12]. We believe it is also suggested that in the $\chi^{(2)}$ ZGP waveguide with only 10 mm in length and multi-wavelength-scale waveguide structure, the MIR output has similar coherence and pulsing characters, as measured in Fig. R4. The technique of electro-optic sampling together with a few-cycle reference pulse could be useful for the accurate temporal characterization of the LWIR output from the $\chi^{(2)}$ ZGP waveguide.

Fig. R3. Characterization of $\chi^{(3)}$ effect in the 10-mm-long $\chi^{(2)}$ ZGP waveguide. (a) Comparison of the measured pump spectra before entering and after propagating through the $\chi^{(2)}$ ZGP waveguide. (b) Comparison of the simulated idler spectra with and without $\chi^{(3)}$ effects.

Fig. R4. The temporal characterization by reproducing the OPG experiment in a bulk ZGP crystal with the same experimental condition of the $\chi^{(2)}$ ZGP waveguide. (a) The measured (black) and retrieved (red) IAC traces. (b) The retrieved temporal profile showing 270 fs pulse width and certain pulse splitting.

Discussions on the OPG coherence and $\chi^{(3)}$ effects have been added into the revised manuscript.

Please see the changes in Line 201 - 207 in the revised manuscript: “In addition, it is worthy to note that the repetition of simulations with and without including $\chi^{(3)}$ nonlinearity is conducted, revealing a nearly identical spectral evolution of the OPG process. Hence, it could be concluded that the generated idler wave is dominated by the $\chi^{(2)}$ nonlinearity. The detailed information about $\chi^{(3)}$ nonlinearity in the demonstrated parametric waveguide could be found in Supplementary Note 3. Meanwhile, a qualitative analysis of partial coherence characteristics of the OPG is also demonstrated in Supplementary Note 3, which needs further investigation in future.

”

Please see the changes in Supplementary Note 3, including the following measurements and simulations:

- The temporal profile characterization of OPG in a bulk ZGP crystal.
- The comparison of the incident and transmitted spectra of the 2.4 μm pump pulse of the $\chi^{(2)}$ ZGP waveguide.
- The simulated idler spectra with and without including the $\chi^{(3)}$ nonlinearity.

[R10]. Manzoni, C. et al. Optical-parametric-generation process driven by femtosecond pulses: Timing and carrier-envelope phase properties. *Phys. Rev. A* 79, 033818 (2009).

[R11]. Jankowski, M. et al. Dispersion-engineered $\chi^{(2)}$ nanophotonics: a flexible tool for nonclassical light. *J. Phys. Photonics* 3, 042005 (2021).

[R12]. Ledezma, L. et al. Intense optical parametric amplification in dispersion-engineered nanophotonic lithium niobate waveguides. *Optica* 9, 303-308 (2022).

[R13]. Nam, S. et al. Octave-spanning mid-infrared femtosecond OPA in a ZnGeP_2 pumped by a 2.4 μm Cr: ZnSe chirped-pulse amplifier. *Opt. Express* 28, 32403-32414 (2020).

[R14]. Hinkelmann, M. et al. High-repetition rate, mid-infrared, picosecond pulse generation with μJ -energies based on OPG/OPA schemes in 2- μm -pumped ZnGeP_2 . *Opt. Express* 28, 21499-21508 (2020).

[R15]. Hong, K. et al. Electric-field reconstruction of femtosecond laser pulses from interferometric autocorrelation using an evolutionary algorithm. *Opt. Commun.* 271, 169-177 (2007).

7. Despite the threshold energy is found to be <1 nJ, yet the required peak power is still in $\sim\text{kW}$ level, which means only the pulse pump scheme is feasible. To minimize the whole system in near future for practical applications, the c. w. pump is highly desired. Is it accessible with all possible efforts for now, such as the optimization of waveguide cross section or trying the OPO method through a microring?

Response: We thank the reviewer for the valuable comments. As commented by the reviewer, c. w. pump is highly desired for miniaturization of the whole system and broader applications such as MIR spectroscopy. In the demonstrated ZGP $\chi^{(2)}$ waveguide, the required peak power for OPG is in $\sim\text{kW}$ level. To reduce the requested peak power, shrink the footprint of the setup, and facilitate c. w. pumping, difference-frequency generation (DFG) could be pursued by choosing two c. w. lasers as the pump and signal sources. Benefiting from the high quality of the ZGP crystal used in the $\chi^{(2)}$ waveguide (manufactured by DIEN TECH, YS-ZGP), which exhibits low transmission loss in the near-infrared region compared to that of traditional ZGP crystal as shown in Fig. R5(a) and R5(b), a tunable c. w. DFG across the mid-infrared band in a new ZGP waveguide driven by near-infrared fiber lasers is feasible. Thus, to demonstrate the prospect of ZGP micro-waveguide towards a miniaturized system, a proof-of-concept experiment of c. w. DFG driven by an all-fiber laser source in a newly fabricated ZGP waveguide is successfully performed.

Fig. R5. (a) The comparison of measured transmission spectra between 8-mm-thick YS-ZGP crystal and traditional ZGP crystal with the same thickness, across the wavelength range of 0.5 - 2.5 μm . (b) The measured transmission spectrum of 8-mm-thick YS-ZGP crystal across 0.5 - 12 μm (The Fresnel reflection is not subtracted). The enlargement of marked red frame is plot in Fig. R5(a).

Fig. R6(a) illustrates the experimental setup of c. w. LWIR DFG in ZGP Waveguide. The setup is composed of three parts, including a home-built c. w. tunable random Raman fiber laser (RRFL) at a wavelength around 1350 nm, which is used as DFG pump source, as depicted in Supplementary Note 4, a commercial erbium-doped fiber amplifier (EDFA) seeded by tunable signal-frequency DFB laser (Conquer, KG-TLS-13-P-FA) emitting in the spectral range of 1527-1567 nm as DFG

signal source, and a newly fabricated ZGP waveguide, cut at $\theta = 65.4^\circ$ and $\varphi = 0^\circ$. The RRFL and EDFA sources are combined by a fiber wavelength division multiplexer (WDM) as an all-fiber laser driver for DFG in the ZGP waveguide.

In the DFG stage, the combined tunable RRFL and EDFA sources with WDM (reflection port: 1550 - 1700 nm, pass port: 1450 - 1490 nm) are first collimated with a broadband fiber collimator (FC). A mechanical chopper is placed at the beam path with a chopping frequency of 1 kHz. Subsequently the laser beam is focused with a CaF₂ lens of a focal length of 100 mm into the ZGP waveguide. The beam diameters at focus are estimated as $\sim 29 \mu\text{m}$ and $33 \mu\text{m}$ for the typical pump wavelength at 1360 nm and signal wavelength at 1565 nm, respectively. Since the RRFL and EDFA sources are both nonpolarized, the specific linearly polarized components of each beam are effectively involved in DFG. The combined pump and signal sources are firstly characterized by a spectral analyzer (Yokogawa AQ6370D) and a power meter (Ophir, 3A). The generated c. w. mid-infrared light after ZGP waveguide is collimated by an uncoated ZnSe lens with a 25 mm focal length and then passes through an AR-coated Germanium filter which is used to block the residual pump/signal waves. For the spectral measurement, the mid-infrared light is focused by a ZnSe lens (3 - 12 μm AR-coated) with a 50-mm focal length into a hollow-core MIR fiber (OptoKnowledge HF500MW), and then detected by a grating-scanning monochromator (Zolix Omni- λ 500i) with a liquid nitrogen cooled HgCdTe detector (Judson, DMCT16-De01).

Fig. R6(b) and R6(c) show the tunable pump spectra and signal spectra, respectively. Although the ZGP waveguide has a fixed phase-matching (PM) angle, the tunable DFG radiation can be obtained by simultaneously tuning the wavelengths of pump and signal light to fulfil PM conditions. When the pump and signal wave power is set both at 250 mW through DFG in ZGP waveguide, tunable mid-infrared lasing across 10.28 - 10.58 μm is obtained by changing the pump/signal wavelength, as shown in Fig. R6(d), which indicates that c. w. pumped ZGP waveguide could be a promising platform for integrated tunable mid-infrared laser generation. More advanced coupling methods such as using lensed fiber coupling can be adopted to further improve the compactness of the system.

Fig. R6. Experimental setup of c. w. DFG based on newly fabricated ZGP waveguide and

characterizations of spectra of pump, signal and corresponding LWIR idler waves. **(a)** The schematic of c. w. mid-infrared laser generation through DFG in ZGP waveguide. WDM, wavelength division multiplexer; L, lens; FC, fiber collimator; LPF, long-pass filter. The setup is composed of three parts, including a home-built c. w. tunable random Raman fiber laser (RRFL) at a wavelength around 1350 nm, which is used as DFG pump source, as depicted in Supplementary Note 4, a commercial erbium-doped fiber amplifier (EDFA) seeded by tunable signal-frequency DFB laser (Conquer, KG-TLS-13-P-FA) emitting in the spectral range of 1527 - 1567 nm as DFG signal source, and a newly fabricated ZGP waveguide, cut at $\theta = 65.4^\circ$ and $\varphi = 0^\circ$. The RRFL and EDFA sources are combined by a fiber wavelength division multiplexer (WDM) as an all-fiber laser driver for DFG in the ZGP waveguide. **(b)** Typical pump spectra tuning from 1340 nm to 1370 nm. **(c)** Typical signal spectra tuning from 1550 nm to 1565 nm. **(d)** The measured spectra of generated LWIR lasing tunable from 10.28 μm to 10.58 μm through DFG in the ZGP waveguide.

The proof-of-concept demonstration of c. w. pumped DFG in $\chi^{(2)}$ ZGP waveguide has been added into Discussion part of manuscript and Supplementary Information.

Please see the changes in Supplementary Note 4.

In the revised manuscript:

“It is worthy to mention that although the remarkable parametric conversion efficiency of 74% in the LWIR region is obtained in the demonstrated ZGP micro-waveguide platform, the required peak power to realize OPG is still in $\sim\text{kW}$ level, which means only pulse pump scheme is feasible. To miniaturize the whole system in near future for practical applications, continuous-wave (c. w.) pump is highly desired. To demonstrate the prospect of ZGP micro-waveguide towards integrated photonics, a proof-of-concept experiment of c. w. DFG driven by an all-fiber laser source in another ZGP waveguide cut at $\theta = 65.4^\circ$ and $\varphi = 0^\circ$ is conducted. Fig. 6(a) illustrates the experimental setup of c. w. DFG in ZGP Waveguide. The setup includes a home-built c. w. tunable random Raman fiber laser (RRFL) around the wavelength of 1350 nm serving as the DFG pump (the detailed laser configuration is depicted in Supplementary Note 5), and a commercial erbium-doped fiber amplifier (EDFA) seeded by a tunable signal-frequency DFB laser (Conquer, KG-TLS-13-P-FA) in the wavelength range of 1527 - 1567 nm as the DFG signal source. The RRFL and the EDFA output are combined by a fiber wavelength division multiplexer to form an all-fiber laser driver for DFG in the ZGP waveguide. Although the ZGP waveguide has a fixed PM angle, the tunable DFG radiation can be obtained by simultaneously tuning the wavelengths of pump and signal light to fulfil PM conditions. The typical tunable spectra of the pump and signal laser sources are presented in Fig. 6(b) and (c), respectively. As illustrated in Fig. 6(d), when the power of pump and signal light is set both as 250 mW, through DFG in ZGP waveguide, tunable LWIR lasing across 10.28 - 10.58 μm can be obtained, which indicates that c. w. pumped ZGP waveguide could be a promising platform for integrated tunable LWIR light generation. More advanced coupling methods such as using lens fiber coupling can be adopted to further improve the compactness of the system. The results also imply that in future, by combining the tunable near-infrared (NIR) single-frequency laser diodes and the developed ZGP micro-waveguide, tunable single-frequency LWIR light can be realized in an integrated waveguide platform, and such an on-chip single-frequency LWIR light source could be a desirable tool for LWIR spectroscopy and free-space communication applications⁵⁰.”

The added new reference in the revised manuscript:

“50. Zou, K. et al. High-capacity free-space optical communications using wavelength- and mode-division-multiplexing in the mid-infrared region. *Nat. Commun.* 13, 7662 (2022).”

We thank the reviewers for the constructive comments of our work and useful suggestions for improving the manuscript. Please find below the response to all the comments.

Reviewer #2

In this paper the authors report on the generation of broadband long wavelength MIR laser by using a ZGP waveguide fabricated by femtosecond laser writing to enhance second-order nonlinear effect. The performance, especially for conversion efficiency, of this waveguide for optical parametric generation is impressive. However, I cannot recommend the publication of this version of manuscript due to the concerns list as below.

1. Although the results and the technical manifestation to be high quality, I find the conceptual novelty to be somewhat lacking. Both physics and tech used in this manuscript including ZGP material, tight light confinement, large mode overlap, wafer bonding and femtosecond laser directing, are well-known in the literature. Less knowledge can be learned from the phenomenon and experimental description. I strongly recommend the authors to seriously summarize the novelty of this work, and focus on it. Such as ZGP waveguide fabrication tech which is the key of high efficiency of OPG.

Response: We thank the reviewer for the careful reading, and helpful suggestions. We agree well with reviewer that the methods employed in our work including ZGP material, tight light confinement, large mode overlap, wafer bonding and femtosecond laser writing are known techniques. However, we would like to stress that in this work employing the existing waveguide fabrication techniques including ultrafast laser direct writing, **we for the first time demonstrate the birefringence crystals-based micro-waveguide for the highly efficient LWIR generation, which generates a quantum conversion efficiency of 74% as a new record in LWIR single-pass parametric processes. The threshold energy is measured as ~ 616 pJ, reduced by more than 1-order of magnitude as compared to those of MIR optical parametric generations in bulk media.** Owing to the inherently efficient quadratic nonlinear response, waveguide structure improved spatial confinement and elongated interaction length, and good coupling efficiency derived from multi-micrometer dimension design in the $\chi^{(2)}$ waveguide with birefringence phase matching, **the extraordinary parametric performance is achieved, which represents a technical breakthrough** in the field of mid-infrared light generation.

In addition, **c. w. all-fiber laser pumped DFG in $\chi^{(2)}$ ZGP waveguide** is demonstrated as the proof-of-concept. Tunable LWIR lasing across 10.28 - 10.58 μm is obtained, which indicates that c. w. pumped ZGP waveguide could be a promising platform for integrated tunable LWIR light generation. More advanced coupling methods such as using lensed fiber coupling can be adopted to further improve the compactness of the system. The combination of $\chi^{(2)}$ waveguide and NIR c. w. fiber lasers or laser diodes provides possibilities to realize on-chip nonlinear frequency converter, which could be a versatile platform for various promising applications in LWIR region such as up-conversion spectroscopy/imaging and quantum photon-pair generation.

Thirdly, we have shown that using ultrafast laser direct writing, the $\chi^{(2)}$ birefringence crystals-based micro-waveguide can be successfully fabricated. Such fabrication method could be extended to various $\chi^{(2)}$ birefringence crystals including ZGP, AgGsS₂, LiGaS₂ and CdSiP₂, **and therefore the LWIR light generation and applications based on $\chi^{(2)}$ birefringence crystals can now be performed on the waveguide platforms.**

We therefore believe that although both physics and techniques are known, the demonstrated $\chi^{(2)}$ micro-waveguide device with birefringence phase matching fabricated by the simple technique would flourish the MIR integrated photonics researches and promote practical applications of MIR spectroscopy and metrology with very compact systems.

We have highlighted and summarized the novelty, and rewritten the abstract and main body in the revised manuscript:

In abstract:

“The realization of compact and efficient broadband mid-infrared (MIR) lasers has enormous impacts in promoting MIR spectroscopy for various important applications. A number of well-designed waveguide platforms have been demonstrated for MIR supercontinuum and frequency comb generations based on cubic nonlinearities, but unfortunately third-order nonlinear response is inherently weak. Here, we propose and demonstrate for the first time a $\chi^{(2)}$ micrometer waveguide platform based on birefringence phase matching for long-wavelength infrared (LWIR) laser generation with a high quantum efficiency. In a ZnGeP₂-based waveguide platform, an octave-spanning spectrum covering 5 - 11 μm is generated through optical parametric generation (OPG). A quantum conversion efficiency of 74% as a new record in LWIR single-pass parametric processes is achieved. The threshold energy is measured as ~ 616 pJ, reduced by more than 1-order of magnitude as compared to those of MIR OPGs in bulk media. Our prototype micro-waveguide platform could be extended to other $\chi^{(2)}$ birefringence crystals and trigger new frontiers of MIR integrated nonlinear photonics.”

In Introduction:

“Up to now, nonlinear frequency conversions have become the main route for generating coherent ultra-broadband MIR radiations, which however generally consist of large and complicated laser apparatuses and exhibit relatively low conversion efficiency^{2,3}.”

“However, $\chi^{(3)}$ nonlinear response is inherently weak. Hence, dedicated dispersion engineering, moderate-to-small effective mode area, and resonators with high quality factors are usually required to achieve a reasonable conversion efficiency^{4-6, 9-11}. On the other hand, more efficient quadratic nonlinearity ($\chi^{(2)}$)-based waveguide devices for parametric conversions such as optical parametric generation/amplification (OPG/OPA), and difference-frequency generation (DFG) are expected to be a promising alternative approach for simple and efficient generation of ultra-broadband MIR lasers.”

“In this work, we, for the first time to the best of our knowledge, propose and demonstrate the $\chi^{(2)}$ parametric micro-waveguide platform with birefringence PM for highly efficient single-pass LWIR generation, taking the advantages of high birefringence $\chi^{(2)}$ in non-oxide nonlinear crystals, such as ZnGeP₂ (ZGP), AgGsS₂, GaSe and CdSiP₂ which are attractive for broadband LWIR generation¹⁶⁻¹⁹.”

“Besides the remarkable laser specifications, the micro-waveguide device fabrication adopts a simple and effective device fabrication technique, namely bonding and grinding the ZGP nonlinear crystal with a designed PM angle, followed by patterning using ultrafast laser direct writing (ULDW)

technique, which provides a simple, universal and effective fabrication method for nonlinear micro-photonics devices. Hence the demonstrated prototype micro-waveguide platforms could be extended to other $\chi^{(2)}$ birefringence crystals for highly efficient and broadband tunable MIR laser generation. This work opens an exciting and simple route towards the portable or on-chip MIR nonlinear photonics and practical applications of MIR spectroscopy and metrology.”

In Discussion:

“It is worthy to mention that although the remarkable parametric conversion efficiency of 74% in the LWIR region is obtained in the demonstrated ZGP micro-waveguide platform, the required peak power to realize OPG is still in \sim kW level, which means only pulse pump scheme is feasible. To miniaturize the whole system in near future for practical applications, continuous-wave (c. w.) pump is highly desired. To demonstrate the prospect of ZGP micro-waveguide towards integrated photonics, a proof-of-concept experiment of c. w. DFG driven by an all-fiber laser source in another ZGP waveguide cut at $\theta = 65.4^\circ$ and $\varphi = 0^\circ$ is conducted. Fig. 6(a) illustrates the experimental setup of c. w. DFG in ZGP Waveguide. The setup includes a home-built c. w. tunable random Raman fiber laser (RRFL) around the wavelength of 1350 nm serving as the DFG pump (the detailed laser configuration is depicted in Supplementary Note 4), and a commercial erbium-doped fiber amplifier (EDFA) seeded by a tunable signal-frequency DFB laser (Conquer, KG-TLS-13-P-FA) in the wavelength range of 1527 - 1567 nm as the DFG signal source. The RRFL and the EDFA output are combined by a fiber wavelength division multiplexer to form an all-fiber laser driver for DFG in the ZGP waveguide. Although the ZGP waveguide has a fixed PM angle, the tunable DFG radiation can be obtained by simultaneously tuning the wavelengths of pump and signal light to fulfil PM conditions. The typical tunable spectra of the pump and signal laser sources are presented in Fig. 6(b) and (c), respectively. As illustrated in Fig. 6(d), when the power of pump and signal light is set both as 250 mW, through DFG in ZGP waveguide, tunable LWIR lasing across 10.28 - 10.58 μ m can be obtained, which indicates that c. w. pumped ZGP waveguide could be a promising platform for integrated tunable LWIR light generation. More advanced coupling methods such as using lens fiber coupling can be adopted to further improve the compactness of the system. The results also imply that in future, by combining the tunable near-infrared (NIR) single-frequency laser diodes and the developed ZGP micro-waveguide, tunable single-frequency LWIR light can be realized in an integrated waveguide platform, and such an on-chip single-frequency LWIR light source could be a desirable tool for LWIR spectroscopy and free-space communication applications⁵⁰.”

Fig. 6 | Experimental setup of c. w. DFG based on another ZGP waveguide and characterizations of spectra of pump, signal, and corresponding LWIR idler waves. (a) The schematic of c. w. mid-infrared laser generation through DFG in ZGP waveguide. WDM, wavelength division multiplexer; L, lens; FC, fiber collimator; LPF, long-pass filter. The setup is composed of three parts, including a home-built c. w. tunable random Raman fiber laser (RRFL) at a wavelength around 1350 nm, which is used as DFG pump source, as depicted in Supplementary Note 4, a commercial erbium-doped fiber amplifier (EDFA) seeded by tunable signal-frequency DFB laser (Conquer, KG-TLS-13-P-FA) emitting in the spectral range of 1527-1567 nm as DFG signal source, and another ZGP waveguide, cut at $\theta = 65.4^\circ$ and $\varphi = 0^\circ$. The RRFL and EDFA sources are combined by a fiber wavelength division multiplexer (WDM) as an all-fiber laser driver for DFG in the ZGP waveguide. (b) Typical pump spectra tuning from 1340 nm to 1370 nm. (c) Typical signal spectra tuning from 1550 nm to 1565 nm. (d) The measured spectra of generated LWIR lasing tunable from 10.28 μm to 10.58 μm through DFG in the ZGP waveguide.

In Conclusion:

“We anticipate that our work opens new possibilities to generate LWIR light sources and perform nonlinear frequency conversion-based applications in LWIR on an integrated photonics platform. First, the waveguide fabrication technique introduced in this work is simple, flexible and universal to various $\chi^{(2)}$ birefringence crystals including AgGsS₂, LiGaS₂ and CdSiP₂, for different parametric conversions and applications. For example, the LiGaS₂ micro-waveguide can enable LWIR parametric conversion pumped by commercially mature 1 μm fs lasers. Second, the demonstrated record high LWIR single-pass parametric quantum efficiency in ZGP waveguide manifests that the $\chi^{(2)}$ nonlinear micro-waveguide platform can lead to a new generation of highly efficient and broadband tunable MIR laser sources, especially in the LWIR region.”

The added new reference in the revised manuscript:

- “4. Griffith, A. G. et al. Silicon-chip mid-infrared frequency comb generation. Nat. Commun. 6, 6299 (2015).
5. Yu, M. et al. Mode-locked mid-infrared frequency combs in a silicon microresonator. Optica 3,

854-860 (2016).

6. Gaeta, A. L., Lipson, M. & Kippenberg, T. J. Photonic-chip-based frequency combs. *Nat. Photonics* 13, 158-169 (2019).

50. Zou, K. et al. High-capacity free-space optical communications using wavelength- and mode-division-multiplexing in the mid-infrared region. *Nat. Commun.* 13, 7662 (2022).”

2. The authors claim “integrated nonlinear photonics”, “integrated waveguide”, “on-chip”, many times in the contents of manuscript. But, in my opinion, the experimental medium is only a planar waveguide, more like a fiber but weakly related to integrated photonics. I do not see in this current realization any path to allow an integrated system, utilizing the large table-top pumping, coupling structure and filters. The authors should provide a detail discussion on the prospects of miniaturization and the performance of such envisioned system in line 214-223.

Response: We thank the reviewer for the comments. The reviewer is correct that the experimental medium is a planar waveguide. We have removed the phrases of “integrated nonlinear photonics”, “integrated waveguide”, and “on-chip”, and modified the related description and rewritten the manuscript. **Moreover, to demonstrate the prospect of ZGP micro-waveguide towards a miniaturized system, a proof-of-concept experiment of c. w. DFG driven by an all-fiber laser source in a newly fabricated ZGP waveguide is successfully performed.**

Fig. R7(a) illustrates the experimental setup of c. w. LWIR DFG in ZGP Waveguide. The setup is composed of three parts, including a home-built c. w. tunable random Raman fiber laser (RRFL) at a wavelength around 1350 nm, which is used as DFG pump source, as depicted in Supplementary Note 4, a commercial erbium-doped fiber amplifier (EDFA) seeded by tunable signal-frequency DFB laser (Conquer, KG-TLS-13-P-FA) emitting in the spectral range of 1527 - 1567 nm as DFG signal source, and a newly fabricated ZGP waveguide, cut at $\theta = 65.4^\circ$ and $\varphi = 0^\circ$. The RRFL and EDFA sources are combined by a fiber wavelength division multiplexer (WDM) as an all-fiber laser driver for DFG in the ZGP waveguide. Although the ZGP waveguide has a fixed PMangle, the tunable DFG radiation can be obtained by simultaneously tuning the wavelengths of pump and signal light to fulfil PM conditions.

In the DFG stage, the combined tunable RRFL and EDFA sources with WDM (reflection port: 1550 - 1700 nm, pass port: 1450 - 1490 nm) are first collimated with a broadband fiber collimator (FC). A mechanical chopper is placed at the beam path with a chopping frequency of 1 kHz. Subsequently the laser beam is focused with a CaF₂ lens of a focal length of 100 mm into the ZGP waveguide. The beam diameters at focus are estimated as $\sim 29 \mu\text{m}$ and $33 \mu\text{m}$ for the typical pump wavelength at 1360 nm and signal wavelength at 1565 nm, respectively. Since the RRFL and EDFA sources are both nonpolarized, the specific linearly polarized components of each beam are effectively involved in DFG. The combined pump and signal sources are firstly characterized by a spectral analyzer (Yokogawa AQ6370D) and a power meter (Ophir, 3A). The generated c. w. mid-infrared light after ZGP waveguide is collimated by an uncoated ZnSe lens with a 25 mm focal length and then passes through an AR-coated Germanium filter which is used to block the residual pump/signal waves. For the spectral measurement, the mid-infrared light is focused by a ZnSe lens (3-12 μm AR-coated) with a 50-mm focal length into a hollow-core MIR fiber (OptoKnowledge HF500MW), and then detected by a grating-scanning monochromator (Zolix Omni- λ 500i) with a liquid nitrogen cooled HgCdTe detector (Judson, DMCT16-De01).

The typical tunable spectra of the pump and signal laser sources are presented in Fig. 6(b) and (c), respectively. In addition, as illustrated in Fig. 6(d), when the power of pump and signal light is set both as 250 mW, through DFG in ZGP waveguide, tunable LWIR lasing across 10.28 - 10.58 μm can be obtained, which indicates that c. w. pumped ZGP waveguide could be a promising platform for integrated tunable LWIR light generation. More advanced coupling methods such as using lensed fiber coupling can be adopted to further improve the compactness of the system.

Fig. R7. Experimental setup of c. w. DFG based on newly fabricated ZGP waveguide and characterizations of spectra of pump, signal, and corresponding LWIR idler waves. **(a)** The schematic of c. w. mid-infrared laser generation through DFG in ZGP waveguide. WDM, wavelength division multiplexer; L, lens; FC, fiber collimator; LPF, long-pass filter. The setup is composed of three parts, including a home-built c. w. tunable random Raman fiber laser (RRFL) at a wavelength around 1350 nm, which is used as DFG pump source, as depicted in Supplementary Note 4, a commercial erbium-doped fiber amplifier (EDFA) seeded by tunable signal-frequency DFB laser (Conquer, KG-TLS-13-P-FA) emitting in the spectral range of 1527 - 1567 nm as DFG signal source, and a newly fabricated ZGP waveguide, cut at $\theta = 65.4^\circ$ and $\varphi = 0^\circ$. The RRFL and EDFA sources are combined by a fiber wavelength division multiplexer (WDM) as an all-fiber laser driver for DFG in the ZGP waveguide. **(b)** Typical pump spectra tuning from 1340 nm to 1370 nm. **(c)** Typical signal spectra tuning from 1550 nm to 1565 nm. **(d)** The measured spectra of generated LWIR lasing tunable from 10.28 μm to 10.58 μm through DFG in the ZGP waveguide.

The proof-of-concept demonstration of c. w. pumped DFG in $\chi^{(2)}$ ZGP waveguide has been added into Discussion part of manuscript and Supplementary Information.

Please see the changes in Supplementary Note 4.

In the revised manuscript:

“It is worthy to mention that although the remarkable parametric conversion efficiency of 74% in the LWIR region is obtained in the demonstrated ZGP micro-waveguide platform, the required peak

power to realize OPG is still in ~kW level, which means only pulse pump scheme is feasible. To miniaturize the whole system in near future for practical applications, continuous-wave (c. w.) pump is highly desired. To demonstrate the prospect of ZGP micro-waveguide towards integrated photonics, a proof-of-concept experiment of c. w. DFG driven by an all-fiber laser source in another ZGP waveguide cut at $\theta = 65.4^\circ$ and $\varphi = 0^\circ$ is conducted. Fig. 6(a) illustrates the experimental setup of c. w. DFG in ZGP Waveguide. The setup includes a home-built c. w. tunable random Raman fiber laser (RRFL) around the wavelength of 1350 nm serving as the DFG pump (the detailed laser configuration is depicted in Supplementary Note 4), and a commercial erbium-doped fiber amplifier (EDFA) seeded by a tunable signal-frequency DFB laser (Conquer, KG-TLS-13-P-FA) in the wavelength range of 1527 - 1567 nm as the DFG signal source. The RRFL and the EDFA output are combined by a fiber wavelength division multiplexer to form an all-fiber laser driver for DFG in the ZGP waveguide. Although the ZGP waveguide has a fixed PM angle, the tunable DFG radiation can be obtained by simultaneously tuning the wavelengths of pump and signal light to fulfil PM conditions. The typical tunable spectra of the pump and signal laser sources are presented in Fig. 6(b) and (c), respectively. As illustrated in Fig. 6(d), when the power of pump and signal light is set both as 250 mW, through DFG in ZGP waveguide, tunable LWIR lasing across 10.28 - 10.58 μm can be obtained, which indicates that c. w. pumped ZGP waveguide could be a promising platform for integrated tunable LWIR light generation. More advanced coupling methods such as using lens fiber coupling can be adopted to further improve the compactness of the system. The results also imply that in future, by combining the tunable near-infrared (NIR) single-frequency laser diodes and the developed ZGP micro-waveguide, tunable single-frequency LWIR light can be realized in an integrated waveguide platform, and such an on-chip single-frequency LWIR light source could be a desirable tool for LWIR spectroscopy and free-space communication applications⁵⁰.”

The added new reference in the revised manuscript:

“50. Zou, K. et al. High-capacity free-space optical communications using wavelength- and mode-division-multiplexing in the mid-infrared region. *Nat. Commun.* 13, 7662 (2022).”

3. Why choosing 10 mm length? What’s the limitation?

Response: We thank the reviewer for the question. The 10 mm length of the $\chi^{(2)}$ ZGP waveguide is chosen based on the calculation of the temporal walk off and parametric back conversion. In our work, the pump pulse at 2.4 μm has a pulse width of 320 fs, and the group velocity mismatch (GVM) between the 2.4 μm pump and 8 μm idler is calculated to be 30 fs/mm, which indicates that the maximum ZGP waveguide length before the significant temporal walk-off happening is 10.7 mm. In addition, simulation of the OPG process in the $\chi^{(2)}$ ZGP waveguide is conducted based on a modified coupled-wave equations [R16, R17]. With a pump pulse energy of 9 nJ which is at the high pumping level, as shown in Fig. R8, parametric back conversion occurs in the ZGP waveguide with a length of ~ 10 mm. (It is worth noting that in linear scale, the signal and idler waves become observable only after a waveguide length of 8 mm, as the power before 8 mm is very weak.) Thus 10 mm length of the $\chi^{(2)}$ ZGP waveguide is chosen. Moreover, we also ensure that propagation in the 10-mm-long ZGP waveguide does not alter the temporal profile of the pump pulse by either the accumulated chirp or nonlinearity such as self-phase modulation as measured in Fig R9.

Fig. R8. The simulated OPG process in the temporal and spectral domains as a function of the ZGP waveguide length, revealing back conversion happening at a length of ~ 10 mm.

Fig. R9. Comparison of the measured pump spectra before entering and after propagating through the 10-mm-long $\chi^{(2)}$ ZGP waveguide.

[R16]. Jankowski, M. et al. Dispersion-engineered $\chi^{(2)}$ nanophotonics: a flexible tool for nonclassical light. *J. Phys. Photonics* 3, 042005 (2021).

[R17]. Ledezma, L. et al. Intense optical parametric amplification in dispersion-engineered nanophotonic lithium niobate waveguides. *Optica* 9, 303-308 (2022).

4. In line 97, the authors claim the simulation loss of the waveguide is less than 0.25 dB/cm, how about real fabricated devices. I suggest giving a detail characterization of samples. In line 125, the roughness of the waveguide is 10 μm , which is compatible with processing wavelength, it's really not good for a low loss waveguide. If the roughness is 10 μm , the mode index changing rapidly, along waveguide. How about the influence for phase matching and other

effects.

Response: We thank the reviewer for the valuable question and suggestion. A detail characterization of the $\chi^{(2)}$ ZGP waveguide, especially the surface roughness and transmission loss are conducted. Cut-back measurement technique [R18] is used for waveguide transmission loss calibration. Two waveguide samples with different length of 8 mm and 10 mm are employed. The input pump wave (2.4 μm) power is set as 60 mW. The averaged insertion losses of the 10 mm and 8 mm waveguides are measured as 10.24 dB and 10.08 dB, respectively, obtained by five repeated measurements. Thus, the transmission loss of the $\chi^{(2)}$ ZGP waveguide at the 2.4 μm pump wavelength in TM polarization is measured as 0.8 dB/cm. Similar measurement process is executed for the signal wave at a wavelength of 3.4 μm in TM polarization, which shows a transmission loss of 1.2 dB/cm. Notably, The discrepancy between the measured and simulated transmission loss is attributed to the wave scattering by the waveguide surface and side walls.

In addition, a 3-dimensional characterization of the ZGP waveguide by using the laser microscope (Olympus, OSL5000) is conducted. The measurement zone area is $\sim 643 \times 644 \mu\text{m}^2$ with a 20 \times objective lens. As shown in Fig.R10 (a, c), the width and depth of the groove is measured as 30.5 μm and 54.4 μm , respectively. To accurately characterize the microscopic profile of the groove sidewall, the ZGP waveguide is tilted by 30 degrees (Fig. R10 (b, d)). In the orientation parallel to the micro-grooves, the line roughness of the side wall is measured as 0.596 μm . Meanwhile, an average line roughness of 0.595 μm is obtained by measuring eight grooves, and the standard deviation is calculated as 0.051 μm . We therefore suggest that with a surface roughness of $< 0.6 \mu\text{m}$, the waveguide scattering loss could be relatively small, and no significant influence is imposed on the waveguide mode and phase-matching condition. We apologize for the inaccurate estimation in the original manuscript. The measure result is added in the revised manuscript and Supplement.

Fig. R10. The 3-dimensional characterization of the waveguide surface and side-wall roughness. **(a)** The scanning-electron image of the $\chi^{(2)}$ ZGP waveguide. **(b)** The measurement apparatus: laser microscope (Olympus, OSL5000), and the schematic of the microscopic profile measurement of the groove sidewall by tilting the ZGP waveguide by 30 degrees. **(c)** The measured cross-sectional profile of the $\chi^{(2)}$ ZGP waveguide. **(d)** The measured three-dimensional surface profile of the $\chi^{(2)}$ ZGP waveguide. An average line roughness between points A and B on the micro-groove side wall by measuring eight grooves, revealing an averaged roughness value of 0.595 μm . The baseline of 0 μm is set based on the original surface of the ZGP crystal. The sampling length is 639.4 μm .

The cut-back measurement of the waveguide transmission loss and the 3-dimensional characterization of the waveguide surface roughness have been added into the revised manuscript and Supplementary Information. Corresponding modification in the revised manuscript is edit.

Please see the changes in Supplementary Note 5 and revised manuscript.

In Line 142 - 145 in the revised manuscript:

“It is measured that the structured ZGP waveguide exhibits a side-wall roughness is less than 0.6 μm (see detail information in the Supplementary Note 5), and the waveguide facets are nearly identical to those of unprocessed ZGP films, indicating a small scattering loss and potentially good coupling efficiency of the fabricated $\chi^{(2)}$ waveguide. In addition, the transmission loss of fabricated waveguide at 2.4 μm pump wavelength in TM polarization is also measured to be 0.8 dB/cm (Supplementary Note 5), indicating a good waveguide performance for the short nonlinear devices.”

[R18]. Hunsperger, R. G. Integrated Optics: Theory and Technology (New York: Springer, 2009).

5. The experimental details and results need to be claimed clearly. Line 166, how can get the gain of waveguide is 60 dB/cm?

Response: We thank the reviewer for the question. For the demonstrated OPG, the measured maximum and minimum idler power are 0.83 mW and 1.1 nW, respectively, which means the output idler power increases by 58.6 dB. Notably, the measured minimum idler power is limited by the noise floor of the MCT detector (defined as 0.3 nW) and the 1-s integration time used for the lock-in detection. Hence, in this work, the parametric gain of OPG in $\chi^{(2)}$ ZGP waveguide should be larger than 58.6 dB/cm. (The fitted OPG gain could be found in Ref. 12 in the revised manuscript.) That is the reason we wrote OPG gain > 60 dB in the original manuscript. However, to avoid confusion, the parametric gain of 58.6 dB/cm is adopted in the revised manuscript.

We thank the reviewers for the constructive comments of our work and useful suggestions for improving the manuscript. Please find below the response to all the comments.

Reviewer #3

Hu et al. presents experimental results on OPG using critical phase-matching in a 10 mm long, large-area ZGP waveguide pumped with 320-fs pulse at 2.4 μm and producing an octave-spanning idler around 8 μm . The use of critical phase matching on a waveguide structure is achieved by misaligning the waveguide with respect to the crystal axes while using TE mode for the pump, and TM modes for the signal/idler. The device development and the presented results are significant, especially with respect to the spectral coverage, threshold, and efficiency. However, the paper needs substantial attention in terms of the accuracy of several claims and statements as well as better positioning of the work with respect to state of the art. Here are some detailed comments:

1. While the authors cite other integrated nonlinear photonic platforms, such as recent demonstration of OPG in PPLN [11, 12] and OP-GaAs [14], they don't do direct comparisons to the numbers presented in those works. The threshold energies in these other integrated platforms are lower by several orders of magnitude, and while not explicitly claimed in the papers, the quantum efficiencies should also be comparable (if not) higher. Thus, the statement about "a new record in conversion efficiency in MIR" need to be substantiated with a more comprehensive comparison. The current comparison seems to only include bulk crystals.

Response: We thank the reviewer for the good comments. We are sorry for the confusion caused. In our study, we focus on long-wavelength infrared (LWIR) region ($>5 \mu\text{m}$). The demonstration of OPG in PPLN [12, 13] exhibits outstanding output characteristics, but has a spectral range of 1.7 to 2.7 μm , limited by the transparency window limitation of PPLN crystal. For the OPG in OP-GaAs [15], the idler wavelength spans from 9 to 12 μm , however, threshold energy and quantum conversion efficiency are not reported. So we could only compare the OPG conversion efficiency of the demonstrated $\chi^{(2)}$ birefringence waveguide with those of bulk crystals. More explicit claims have been added in the revised manuscript to remove the confusion. In addition, the comparison of PPLN waveguide, OP-GaAs waveguide, and the demonstrated $\chi^{(2)}$ birefringence waveguide has been added into the revised Supplementary Information, as shown in Table R2. It reveals that a good performance especially remarkable quantum efficiency is demonstrated operating in the LWIR region.

Table R2 The comparison of PPLN waveguide, OP-GaAs waveguide, and the demonstrated $\chi^{(2)}$ birefringence waveguide

Crystal platform	Phase-matching mechanism	OPG wavelength (μm)	Length (mm)	Waveguide width (μm)	Threshold energy (pJ)	Quantum conversion efficiency (%)	Reference
PPLN	Quasi-phase matching	1.7-2.7 ^a	6	1.85	0.06	22.2%	12
OP-	Quasi-phase	9-12 ^b	14.5	10	NA	N.A.	15

GaAs	matching						
ZGP	Birefringence phase matching	7-9 ^a 5-11 ^c	10	40	616	74%	This work

^a idler wavelength range at -10 dB level, ^b idler wavelength range at full width, and ^c idler wavelength range at -30 dB level.

Please see the changes in Discuss part in the revised manuscript:

“In addition, to further reveal the technical breakthrough of the demonstrated ZGP micro-waveguide, the performance comparison with reported PPLN nano-waveguide¹², and OP-GaAs waveguide¹⁵ is also executed (Supplementary Note 6). The OPG in the PPLN nano-waveguide¹² exhibits extraordinary threshold energy, but a spectral range of 1.7 to 2.7 μm , limited by the transparency window of the PPLN crystal, and a relatively low quantum conversion efficiency (22.2%). Based on OP-GaAs crystal, as an attractive MIR nonlinear optical material, it presents an OPG output with the spectrum extending to LWIR region, which is a significant breakthrough in the field of MIR integrated photonics. However, highly efficient single-pass parametric conversion has not been achieved. Notably, although ZGP micro-waveguide-based OPG has a relatively high pump threshold compared to that of PPLN nano-waveguide, benefited from waveguide-enhanced parametric interaction and inherently strong quadratic nonlinearity at the same level with that of OP-GaAs waveguide platform, a remarkable parametric quantum conversion efficiency together with the broadband and tunable LWIR spectrum across 5 - 12 μm is successfully and firstly realized. These overviews and comparisons of MIR single-pass parametric sources show the demonstrated quadratic nonlinear waveguide as a marked advancement of compact and efficient MIR laser sources.”

Please also see the comparison of PPLN waveguide, OP-GaAs waveguide, and the demonstrated $\chi^{(2)}$ birefringence waveguide, as well as Table R2 added in the revised Supplementary Information.

2. In the presented comparisons, the authors state that they are comparing to “state-of-the-art parametric processes.” However, they are only including single-pass parametric processes. Using bulk OPOs, similarly high mid-IR conversion efficiencies and low thresholds have been achieved. This should be either include in the comparison or the statement needs to be revised.

Response: We thank the reviewer for the good comment. We have revised the statement by only comparing the single -pass parametric process in the revised manuscript.

Please see the changes in the revised manuscript such as:

In abstract: “**A quantum conversion efficiency of 74% as a new record in LWIR single-pass parametric process is achieved.**”

In Line 63 - 65: “In this work, we, for the first time to the best of our knowledge, propose and demonstrate the $\chi^{(2)}$ parametric micro-waveguide platform with birefringence PM for highly efficient single-pass LWIR generation,”

And

In Discussion part: “Over the last decade, LWIR single-pass parametric sources have been demonstrated in bulk nonlinear crystals facilitated either by birefringence or quasi-PM techniques.”

3. The authors mention in their abstract that $\chi^{(3)}$ has an inherently low conversion efficiency, and they expand on their introduction. While the strength of nonlinearity is known to be low for $\chi^{(3)}$ processes, the conversion efficiency, however, is not necessarily low. Examples of $\chi^{(3)}$ processes with higher efficiencies include recent demonstrations of Kerr combs using pulsed pumps and dark soliton generation. Furthermore, the supercontinuum processes they cite based on $\chi^{(3)}$ interactions in supercontinuum generation and don't seem to face an issue in terms of conversion efficiency. Such comparisons with $\chi^{(3)}$ can be more accurate.

Response: We thank the reviewer for the valuable comments. The reviewer is correct that the conversion efficiency of $\chi^{(3)}$ process is not necessarily low. We are sorry for the inaccurate statement, and we have corrected it by only mentioning the $\chi^{(3)}$ nonlinearity is inherently weak and further discussion related to $\chi^{(3)}$ process is also added in the revised manuscript. It is worth mentioning that as the strength of $\chi^{(3)}$ nonlinearity is inherently weak, to achieve a high conversion efficiency in the $\chi^{(3)}$ based Kerr combs or dark soliton generally requires high-Q resonators for resonant enhancement. On the other hand, the supercontinuum generation based on $\chi^{(3)}$ interactions needs dedicated dispersion engineering and moderate effective mode field, which increases the design and experimental constrains. On the contrary, in the demonstrated $\chi^{(2)}$ birefringence waveguide, neither high-Q resonators nor dedicated dispersion engineering is needed, while a high parametric conversion efficiency could be realized.

Please see the changes in Line 42 - 44 in the revised manuscript:

“However, $\chi^{(3)}$ nonlinear response is inherently weak. Hence, necessitating dedicated dispersion engineering, moderate effective mode field, and resonators with high quality factors are usually required to achieve a reasonable conversion efficiency^{4-6, 9-11}.”

The added new reference in the revised manuscript:

“4. Griffith, A. G. et al. Silicon-chip mid-infrared frequency comb generation. Nat. Commun. 6, 6299 (2015).

5. Yu, M. et al. Mode-locked mid-infrared frequency combs in a silicon microresonator. Optica 3, 854-860 (2016).

6. Gaeta, A. L., Lipson, M. & Kippenberg, T. J. Photonic-chip-based frequency combs. Nat. Photonics 13, 158-169 (2019)”

4. It seems, from Figure. 2(a), that the optical axis of the crystal is in plane. Therefore, the TM modes should be “ordinary” since they are polarized orthogonal to the optical axis. The TE pump is then extraordinary. The authors make the opposite statement in line 91. Is this a typo? Or is Fig 2a not clear regarding the orientation of the optical axis?

Response: We thank the reviewer for the good comments. We are sorry for the typo, and we have corrected the corresponding description in line 91 in the revised manuscript.

Please see the changes in Line 101 - 102 in the revised manuscript:

“As illustrated in simulated mode cross sections in Fig. 1(b), fundamental mode profiles of the 2.4 μm pump and 8 μm idler in transverse magnetic (TM_{00}) and electric (TE_{00}) polarizations, corresponding to ordinary and extraordinary waves, respectively, of ZGP crystal are well confined and overlapped in the $\chi^{(2)}$ waveguide.”

5. The modes of anisotropic waveguides can be considerably lossy when propagation happens in a direction not aligned with the crystal axes [J. Opt. Soc. Am. A 10, 246-261 (1993)]. The results of this work suggest this is not the case here. Could the authors explain why?

Response: We thank the reviewer for the valuable question. As mentioned by the reviewer, the research work related to [J. Opt. Soc. Am 10, 246-261 (1993)] reveals that the modes of anisotropic waveguides can be considerably lossy when the light propagating along a direction not aligned with optical axes of crystal. However, it is worth noting that the involved theoretical modal is only valid for a weak guiding by the waveguide in which $\Delta_o, \Delta_e \ll 1$ needs to be satisfied, where $\Delta_i = (1 - n_{i,cladding}^2/n_{i,core}^2)/2$, $i = o, e$ represents the index contrast between the waveguide core and cladding. In our work, for the pump wave at a central wavelength of 2.4 μm , $n_{o, core}=3.138$, $n_{e, core}=3.177$, $n_{o, cladding}=n_{e, cladding}=1$ (air), $n_{o, substrate}=n_{e, substrate}=1.43$ (fused silica) are calculated. The corresponding values of Δ_o and Δ_e are then obtained as 0.449 and 0.450, respectively, which does not accord with the weak guiding prerequisite. Since the upper and lower cladding is air and fused silica respectively, which are isotropic and with refractive indices much smaller than that of ZGP core, it can be expected the fundamental modes of the waveguide are not leaky. As a verification, finite-element formulation method [R19] is employed to execute a rigorous numerical electromagnetic analysis. Transmission losses of ordinary and extraordinary fundamental modes at the pump (2.4 μm), signal (3.4 μm) and idler (8 μm) wavelengths when propagating along different angles θ with the optical axis are calculated, as shown in Fig. R11. The loss profiles agree well with the results of [R19] (Fig. 14 (b)). It is seen that the propagation losses are almost independent of the propagation angles with respect to the optical axis, and the loss values of the pump, signal and idler waves are confined to 10^{-7} dB/cm, 10^{-5} dB/cm and 10^{-1} dB/cm which exhibits a reasonably low mode leaky loss in the ZGP $\chi^{(2)}$ birefringence waveguide.

Fig. R11. The numerical analysis based on finite-element method in calculating transmission losses of ordinary and extraordinary fundamental modes at the pump (2.4 μm), signal (3.4 μm) and idler (8 μm) wavelengths when propagating along different angles θ with the optical axis, in the ZGP $\chi^{(2)}$ birefringence waveguide

However, taking the anisotropy characteristics of waveguide and substrate absorption into consideration, the waveguide presents a substantial mode transmission loss resulted from mode leak and material absorption, particularly at wavelength larger than 9 μm . Propagation losses of fundamental modes in the designed $\chi^{(2)}$ waveguide in TE and TM polarizations with the angle between the propagation direction and the crystal optical axis as 48.3° have been recalculated and replotted in Fig. 1(c) in the revised manuscript. Please see changes of Fig. 1(c) in the revised manuscript (also as following). The propagation loss of fundamental TE and TM mode is less than 1.5 dB/cm and 4.0 dB/cm, respectively. Notably, calculated propagation losses of fundamental modes are < 0.05 dB/cm at 8 μm , and < 0.25 dB/cm (TE) and 0.5 dB/cm (TM) at 9 μm , respectively. The raise of waveguide losses in the wavelength range of 9 - 11 μm is attributed to the absorption of fused silica substrate peaked at 9.5 μm , as shown in the top inset which depicts the absorption coefficient (imaginary part of complex refractive index) of silica, as a function of wavelength²². This confirms that the transmission loss in the 10-mm-long ZGP $\chi^{(2)}$ birefringence waveguide is relatively small.

Hence, thanks to the reviewer's comments, we have corrected the Fig. 1(c), modified corresponding description, and added [J. Opt. Soc. Am. A 10, 246-261 (1993)] as a reference in the revised manuscript.

In the revised manuscript:

“Moreover, the propagation loss of fundamental modes including the material absorption and anisotropy related mode leakage^{20, 21} in the designed $\chi^{(2)}$ waveguide in a wavelength range of 2 to 11 μm are calculated, as presented in Fig. 1(c). The propagation loss of fundamental TE and TM mode is less than 1.5 dB/cm and 4.0 dB/cm, respectively. Notably, it is revealed that the pump wave at 2.4 μm in TM_{00} mode has a propagation loss < 0.01 dB/cm, and losses of fundamental modes are < 0.05 dB/cm at 8 μm , and < 0.25 dB/cm (TE) and 0.5 dB/cm (TM) at 9 μm , respectively, which guarantees a good transmission of pump and parametric waves in the $\chi^{(2)}$ waveguide. The raise of waveguide losses in the wavelength range of 9 - 11 μm is attributed to the absorption of fused silica substrate peaked at 9.5 μm , as shown in the top inset which depicts the absorption coefficient of silica, as a function of wavelength²². This implies that substrates such as sapphire with low absorption in the LWIR region could further improve the performance of the nonlinear waveguide.”

Fig. 1(c) Calculated propagation losses of fundamental modes in the designed $\chi^{(2)}$ waveguide in TE and TM polarizations with the angle between the propagation direction and the crystal optical axis as 48.3° , in a broad spectral range of 2 - 11 μm . The results indicate that the propagation loss of fundamental TE and TM mode is less than 1.5 dB/cm and 4.0 dB/cm, respectively. Notably, calculated propagation losses of fundamental modes are less than 0.05 dB/cm at 8 μm , and less than 0.25 dB/cm (TE) and 0.5 dB/cm (TM) at 9 μm , respectively. The raise of waveguide losses in the wavelength range of 9 - 11 μm is attributed to the absorption of fused silica substrate peaked at 9.5 μm , as shown in the top inset which depicts the absorption coefficient (imaginary part of complex refractive index) of silica, as a function of wavelength²². Meanwhile, the material loss of ZGP crystal is minimal in the spectral range of 2 - 11 μm as shown by the measured transmission spectrum of a 10-mm-thick uncoated ZGP crystal in the bottom inset of Fig. 1(c).

The added new reference in the revised manuscript:

“20. Lu, M. & Fejer, M. M. Anisotropic dielectric waveguides. *J. Opt. Soc. Am. A* 10, 246-261 (1993).

21. Liu, H. H. & Chang, H. C. Solving leaky modes on a dielectric slab waveguide involving materials of arbitrary dielectric anisotropy with a finite-element formulation. *J. Opt. Soc. Am. B* 31, (1360-1376) 2014.”

[R19]. Liu, H. and Chang, H. Solving leaky modes on a dielectric slab waveguide involving materials of arbitrary dielectric anisotropy with a finite-element formulation. *J. Opt. Soc. Am. B* 31, 1360-1376 (2014).

6. The phase-matching mechanism used can be interpreted as modal phase matching. This has been demonstrated before in other platforms, but it has been usually weak. The results of this work are enabled by the large nondiagonal tensor (d_{36}) of ZGP. The paper can benefit from such a generalization and comparison with other materials.

Response: We thank the reviewer for the valuable suggestion. As the reviewer mentioned, the phase-matching mechanism used in our work can be interpreted as modal phase matching, but attributed to fundamental mode phase matching based on birefringence phase matching mechanism, which is different from the modal phase matching related to fundamental modes and high-order modes in previous demonstrated microring or microdisk resonators. Hence, the inherent strong quadratic nonlinearity of ZGP crystal could be utilized. In addition, to further understand the advantages of sampled ZGP waveguide, an overview of AgGaS_2 , GaSe , and CdSiP_2 and other nonlinear crystals, which could also be used for LWIR generation and effectively processed by the ULDW technique as birefringence $\chi^{(2)}$ waveguide platforms, is conducted. In our current work, ZGP is used with reasons summarized as following aspects. Firstly, ZGP has an excellent quadratic nonlinear coefficient ($d_{36} \sim 75$ pm/V) which is higher than those of AgGaS_2 and GaSe , only lower than that of CdSiP_2 ($d_{36} \sim 84.5$ pm/V); however, the transparent window of ZGP ($0.74 \sim 12$ μm) is broader than that of CdSiP_2 ($0.5 \sim 9$ μm). Second, high quality ZGP crystals cut into designed phase-matching angles are easy to be obtained with mature growth technologies. Unlike ZGP, GaSe could be cleaved only along the (001) plane (z-cut, $\theta = 0^\circ$), which restricts its usefulness in birefringence $\chi^{(2)}$ waveguide, although GaSe has a broader transparent window. In addition, ZGP has a good thermal conductivity (35 W/mK) which eliminates any thermal related variations of the fabricated $\chi^{(2)}$ waveguide, in case there is any absorption. (In principle there is no obvious absorption loss from the ZGP waveguide.). With above considerations, we choose ZGP as a typical example of birefringence $\chi^{(2)}$ waveguide.

Besides AgGaS_2 , GaSe , and CdSiP_2 , nonlinear crystals with large bandgap energy, such as LiGaS_2 may also be fabricated through the ULDW technique into a $\chi^{(2)}$ waveguide, which could be pumped at ~ 1 μm wavelength. This would broaden the usefulness and impact of demonstrated birefringence $\chi^{(2)}$ waveguide.

Table R3 summarizes and compares the optical properties of some typical long-wavelength IR nonlinear crystals.

Crystals	Point group	Nonlinear coefficient (pm/V)	Transparent range (μm)	Thermal conductivity (W/mK)	Bandgap (eV)	Reference
ZGP	42m	75	0.74-12	35	2.1	R20
AgGaS ₂	42m	12	0.47-13	1.4	2.7	R21
GaSe	62m	57	0.65-18	2.0	2.1	R22
CdSiP ₂	42m	84.5	0.5-9	13.6	2.45	R23
LiGaS ₂	mm2	5.8	0.32-11.6	5.1	4.15	R24, R25

More descriptions and discussions about the choice of nonlinear crystal are added into the revised manuscript and Supplementary Information.

Please see changes in Line 85 - 88 in the revised manuscript: “The ZGP crystal is chosen as the material platform for its high $\chi^{(2)}$ nonlinearity ($d_{36} \sim 75$ pm/V), mature growth technique, and broad MIR transparent range ($\sim 0.73 - 12 \mu\text{m}$)” in the revised manuscript.

Please see the changes in Supplementary Note 1.

[R20]. Sanchez, D. et al. 7 μm , ultrafast, sub-millijoule-level mid-infrared optical parametric chirped pulse amplifier pumped at 2 μm . *Optica* 3, 147-150 (2017).

[R21]. Migal, E. A. et al. Highly efficient optical parametric amplifier tunable from near-to mid-IR for driving extreme nonlinear optics in solids. *Opt. Lett.* 42, 5218-5221 (2017).

[R22]. Liu, K. et al. Multimicrojoule GaSe-based midinfrared optical parametric amplifier with an ultrabroad idler spectrum covering 4.2 - 16 μm . *Opt. Lett.* 44, 1003-1006 (2019).

[R23]. Lesko, D. M. B. et al. A six-octave optical frequency comb from a scalable few-cycle erbium fibre laser. *Nat. Photonics* 15, 281-286 (2021).

[R24]. Nikogosyan, D.N. *Nonlinear optical crystals: a complete survey*, 1st ed. New York, NY: Springer, 2005.

[R25]. Li, W. et al. Theoretical Study on the Intrinsic Source of the Large Thermal Conductivity of Li-Based Chalcogenide Nonlinear Optical Crystals: From AgGaS_2 to LiGaS_2 .

7. The authors claim “dispersion engineering” in line 104, which is not accurate as the waveguide geometry doesn’t seem to affect the overall dispersion. The authors even explicitly mention that in lines 106-108 “...Owing to the multi-wavelength-scale geometry, the effect of waveguide dispersion is weak, and thus the ZGP waveguide features a nearly identical GVD profile with that of the bulk material.....”. The angle they choose cannot be exploited as an independent degree of freedom for dispersion engineering since it's already exhausted by the phase-matching condition. This is a major point of confusion.

Response: We thank the reviewer for the valuable comment. The reviewer is correct that the phrase of “dispersion engineering” is not accurate. In fact, what we intend to express is that benefiting from the multi-wavelength-scale geometry, the designed ZGP waveguide features a nearly identical GVD profile with that of the bulk material. Hence, there is no need of additional considerations when designing the $\chi^{(2)}$ waveguide but just choosing an appropriate phase-matching angle of the bulk crystal, such that the desired parametric conversion could be realized. This simplifies the process of $\chi^{(2)}$ waveguide design. We apologize for the confusion caused, and we have removed the claim of “dispersion engineering”, and added corresponding descriptions of dispersion evaluation of the $\chi^{(2)}$ waveguide in the revised manuscript, to avoid confusion.

Please see the changes in Line 118 - 119 in the revised manuscript:

“Evaluation of dispersion is another critical considerable project for the design of $\chi^{(2)}$ waveguide devices.”

8. The side-wall roughness mentioned on line 125....should it be less than 10 nm as opposed to 10 μm ?

Response: We thank the reviewer for the valuable comment. A detail characterization of the $\chi^{(2)}$ ZGP

waveguide, especially the surface roughness is conducted. A 3-dimensional characterization of the ZGP waveguide by using the laser microscope (Olympus, OSL5000) is conducted. The measurement zone area is $\sim 643 \times 644 \mu\text{m}^2$ with a $20\times$ objective lens. As shown in Fig.R12(a, c), the width and depth of the groove is measured as $30.5 \mu\text{m}$ and $54.4 \mu\text{m}$, respectively. To accurately characterize the microscopic profile of the groove sidewall, the ZGP waveguide is tilted by 30 degrees (Fig. R12(b, d)). Meanwhile, an average line roughness of $0.595 \mu\text{m}$ is obtained by measuring eight grooves, and the standard deviation is calculated as $0.051 \mu\text{m}$. We therefore suggest that with a surface roughness of $< 0.6 \mu\text{m}$, the waveguide scattering loss could be relatively small, and no significant influence is imposed on the waveguide mode and phase-matching condition. We apologize for the inaccurate estimation in the original manuscript. The measure result is added in the revised manuscript and Supplement.

Fig. R12. The 3-dimensional characterization of the waveguide surface and side-wall roughness. **(a)** The scanning-electron image of the $\chi^{(2)}$ ZGP waveguide. **(b)** The measurement apparatus: laser microscope (Olympus, OSL5000), and the schematic of the microscopic profile measurement of the groove sidewall by tilting the ZGP waveguide by 30 degrees. **(c)** The measured cross-sectional profile of the $\chi^{(2)}$ ZGP waveguide. **(d)** The measured three-dimensional surface profile of the $\chi^{(2)}$ ZGP waveguide. An average line roughness between points A and B on the micro-groove side wall by measuring eight grooves, revealing an averaged roughness value of $0.595 \mu\text{m}$. The baseline of $0 \mu\text{m}$ is set based on the original surface of the ZGP crystal. The sampling length is $639.4 \mu\text{m}$.

The 3-dimensional characterization of the waveguide surface roughness have been added into the revised manuscript and Supplementary Information.

Please see the changes in revised manuscript and Supplementary Note 5.

In Line 139 - 142 in the revised manuscript:

“It is measured that the structured ZGP waveguide exhibits a side-wall roughness is less than 0.6 μm (see detail information in the Supplementary Note 5), and the waveguide facets are nearly identical to those of unprocessed ZGP films, indicating a small scattering loss and potentially good coupling efficiency of the fabricated $\chi^{(2)}$ waveguide.”

9. The sentence on the drawbacks of QCLs needs structural revisions.

Response: We thank the reviewer for the comment. We have revised the sentence as “Among various approaches, quantum cascaded lasers show good promise in device compactness; however, quantum cascaded lasers still face limitations in producing ultra-broadband emissions and ultrashort pulses, particularly in the femtosecond time scale.” in the revised manuscript.

Please see the changes in Line 31 - 33 in the revised manuscript.

“Among various approaches, quantum cascaded lasers show good promise in device compactness. However, quantum cascaded lasers still face limitations in producing broadband emissions and ultrashort pulses, particularly in the femtosecond time scale.”

10. “long-wavelength mid-infrared” is probably better changed to mid-to-long-wavelength infrared or something like that. Typically, MWIR and LWIR are distinct, and the source seems to span both.

Response: We thank the reviewer for the good comment. “Long-wavelength mid-infrared” is changed to LWIR in the revised manuscript, as the most focused spectral range is 6 - 12 μm .

Please see the changes in Line 20 - 22 in the revised manuscript:

“Here, we propose and demonstrate for the first time a $\chi^{(2)}$ waveguide platform based on birefringence phase matching for long-wavelength infrared laser (LWIR) generation with a high quantum efficiency.”

11. Measurement of the input coupling using a pinhole approximating the size of their mode seems to neglect the effects of AR coatings, mode mismatch, scattering, etc. First, this method needs to be justified. Second, how the accuracy of such a measurement can affect the claimed numbers should be discussed in more depth.

Response: We thank the reviewer for the valuable comment. To justify the measurement of the input coupling by using a pinhole, we adopt the insertion loss technique to evaluate the coupling loss. At the pump wavelength of 2.4 μm with the TE polarization, the insertion loss which is consisted of the coupling loss, Fresnel loss, and waveguide scattering loss is measured. When an input power of 60 mW is used, an output power of 5.67 mW is obtained with an insertion loss measured as 10.24 dB. The Fresnel reflection loss is calculated to be 1.36 dB. As the response to the 4th comment of the 2nd reviewer, cut-back measurement technique [R26] is used for waveguide scattering loss calibration. Two waveguide samples with different length of 8 mm and 10 mm are employed. The input pump wave (2.4 μm) power is set as 60 mW. The averaged losses of the 10 mm and 8 mm waveguides are measured as 10.24 dB and 10.08 dB, respectively, obtained by five repeated measurements. Thus, the propagation loss including scattering loss of the $\chi^{(2)}$ ZGP waveguide at the 2.4 μm pump wavelength in TE polarization is obtained as 0.8 dB/cm. Notably, The discrepancy between the measured and simulated transmission loss is attributed to the wave scattering by the

waveguide surface and side walls. Thus the coupling loss is calculated to be $10.24 - 1.36 - 0.8 = 8.08$ dB. Therefore, we could conclude that the coupling efficiency is 15.6%, which is similar to the value (15.3%) obtained by the pinhole measurement method. It is worth mentioning that as in the insertion loss measurement, both the fundamental and higher-order modes output are collected, the mode-mismatch loss is not counted, and the coupling efficiency includes those coupled to both the fundamental and higher-order modes in the $\chi^{(2)}$ ZGP waveguide.

To evaluate the accuracy of pinhole measurement, repeated measurements for 6 times are conducted with the same input power of the 2.4 μm pump wave. As shown in Fig. R13, the measured maximum and minimum coupling efficiencies are 14.2% and 16.7%, respectively, corresponding to quantum conversion efficiencies in the range of 67.8% to 79%. An average quantum conversion efficiency is obtained as 73%, which is close to our claimed number. Hence, considering the above all, we believe that pinhole measurement method is relatively accurate.

Fig. R13. The repeated coupling efficiency measurement for six times by using the pinhole method, and the corresponding quantum conversion efficiencies.

The calibration of the coupling efficiency by using the insertion loss measurement and the repeated pinhole measurement have been added into Supplementary Information.

Please see the changes in Supplementary Note 7.

[R26]. Hunsperger, R. G. Integrated Optics: Theory and Technology (New York: Springer, 2009).

REVIEWERS' COMMENTS

Reviewer #2 (Remarks to the Author):

All my concern has been treated & revised, I think now it can be accepted for publication.

Reviewer #4 (Remarks to the Author):

The authors have carefully revised the manuscript and addressed all my concerns systematically. I am happy to recommend publication.

We thank the reviewers for the constructive comments of our work and useful suggestions for improving the manuscript. Please find below the response to all the comments.

Reviewer #1 All my concern has been treated & revised, I think now it can be accepted for publication.

Response: We thank the referee again for recommending the publication of our manuscript.

Reviewer #2 The authors have carefully revised the manuscript and addressed all my concerns systematically. I am happy to recommend publication.

Response: We thank the referee again for recommending the publication of our manuscript.